# Altered DNA methylation within DNMT3A, AHRR, LTA/TNF loci mediates the effect of smoking on inflammatory bowel disease

Han Zhang [1,12], Rahul Kalla [2,12], Jie Chen [1,12], Jianhui Zhao[1], Xuan Zhou [1,3], Alex Adams[4], Alexandra Noble[4], Nicholas T. Ventham [5], Judith Wellens [4,6], Gwo-Tzer Ho[2], Malcolm G. Dunlop [7], Jan Krzysztof Nowak[8], Yuan Ding[9], Zhanju Liu [10,13] ✉, Jack Satsangi [4,13] ✉, Evropi Theodoratou [7,11,13] ✉ & Xue Li [1,3,13] ✉

This work aims to investigate how smoking exerts effect on the development of inflammatory bowel disease (IBD). A prospective cohort study and a Mendelian randomization study are first conducted to evaluate the association between smoking behaviors, smoking-related DNA methylation and the risks of Crohn's disease (CD) and ulcerative colitis (UC). We then perform both genome-wide methylation analysis and co-localization analysis to validate the observed associations. Compared to never smoking, current and previous smoking habits are associated with increased CD ($P = 7.09 \times 10^{-10}$) and UC ($P < 2 \times 10^{-16}$) risk, respectively. DNA methylation alteration at cg17742416 [*DNMT3A*] is linked to both CD ($P = 7.30 \times 10^{-8}$) and UC ($P = 1.04 \times 10^{-4}$) risk, while cg03599224 [*LTA/TNF*] is associated with CD risk ($P = 1.91 \times 10^{-6}$), and cg14647125 [*AHRR*] and cg23916896 [*AHRR*] are linked to UC risk ($P = 0.001$ and 0.002, respectively). Our study identifies biological mechanisms and pathways involved in the effects of smoking on the pathogenesis of IBD.

Inflammatory bowel diseases (IBD) are chronic and recurrent inflammatory disorders affecting the digestive tract, which consist of two prominent subtypes: ulcerative colitis (UC) and Crohn's disease (CD). The incidence and prevalence of IBD continue to rise globally, with recent estimates suggesting that between 0.5 - 1% of the Europeans suffer from the disease[1]. Epidemiological evidence from migration and other studies increasingly implicate the exposome in IBD development[2]. Among environmental triggers, cigarette smoking has been the most intensively investigated. Cigarette smoking has repeatedly been linked to IBD risk in observational studies. The reported differential effects of smoking status (i.e., current smoking and previous smoking) on UC and CD are divergent[3,4]. Despite decades

[1]Department of Big Data in Health Science School of Public Health and The Second Affiliated Hospital, Zhejiang University School of Medicine, Hangzhou, Zhejiang, China. [2]Edinburgh IBD Science Unit, Centre for Inflammation Research, University of Edinburgh, Edinburgh, UK. [3]Centre for Population Health Sciences, Usher Institute, University of Edinburgh, Edinburgh, UK. [4]Translational Gastroenterology Unit, Nuffield Department of Medicine, Experimental Medicine Division, University of Oxford, John Radcliffe Hospital, Oxford, UK. [5]Academic Coloproctology, University of Edinburgh, Western General Hospital, Edinburgh, UK. [6]Department of Chronic Diseases and Metabolism, Translational Research Center for Gastrointestinal Disorders (TARGID), KU Leuven, Leuven, Belgium. [7]Cancer Research UK Scotland Centre and Medical Research Council Human Genetics Unit, University of Edinburgh, Edinburgh, UK. [8]Department of Paediatric Gastroenterology and Metabolic Diseases, Poznan University of Medical Sciences, Poznan, Poland. [9]Department of Hepatobiliary and Pancreatic Surgery, the Second Affiliated Hospital, Zhejiang University School of Medicine, Hangzhou, China. [10]Center for IBD Research, Shanghai Tenth People's Hospital, Tongji University School of Medicine, Shanghai, China. [11]Centre for Global Health, Usher Institute, University of Edinburgh, Edinburgh, UK. [12]These authors contributed equally: Han Zhang, Rahul Kalla, and Jie Chen. [13]These authors jointly supervised this work: Zhanju Liu, Jack Satsangi, Evropi Theodoratou, Xue Li. ✉e-mail: liuzhanju88@126.com; jack.satsangi@ndm.ox.ac.uk; e.theodoratou@ed.ac.uk; xueli157@zju.edu.cn

of research using different methodologies, our current understanding of the mechanisms underlying these effects remains limited[5], partly owing to inherent limitations associated with observational studies, including residual confounding and reverse causation.

To overcome these limitations in an era when genome-wide association studies (GWAS) data is available, Mendelian Randomization (MR) analysis, which leverages genetic variants as proxies, is now widely applied to evaluate causal associations between modifiable exposures and disease-related traits[6]. Since genetic variables are assorted randomly at conception before the occurrence of diseases, they are immune to self-adapted lifestyle and environmental factors. Furthermore, with the development of epigenome-wide association studies (EWAS), tobacco smoking has been shown to infer DNA methylation status changes at an epigenome-wide level[7–9]. In parallel, DNA methylation alterations have also been implicated in the underlying pathogenesis of IBD[10–12]. It has been proposed that smoking-related DNA methylation changes are mechanistically involved in CD risk in a case-control study, but there are relatively few relevant data[13]. We hypothesize that tobacco smoking may affect the susceptibility of IBD by modulating DNA methylation status in the cells of the circulating immune system.

In the current work, we aim first to assess the association between cigarette smoking and the risk of IBD through a prospective cohort study in the UK Biobank. Then, we evaluate the potential causal association between smoking behaviors and smoking-related DNA methylation with IBD by performing MR analysis, and finally validate the associations via genome-wide methylation and colocalization analyzes.

## Results

### Smoking and IBD risk

The demographic profiles of the subjects are presented in Supplementary Table 1. Compared with never smoking, the hazard ratios (HRs) and 95% confidence interval (CI) of CD for ever smoking, previous smoking, and current smoking in the fully adjusted model were 1.49 (95%CI: 1.31–1.69, $P = 5.83 \times 10^{-10}$), 1.41 (95%CI: 1.23–1.61, $P = 7.00 \times 10^{-7}$), and 1.81 (95%CI: 1.50–2.19, $P = 7.09 \times 10^{-10}$), respectively (Table 1). Stratified analysis replicated these positive associations in those over 60 years old (all $P < 0.05$), but not in those under 60 (all $P > 0.05$, Table 2). The HRs and 95%CI for UC in the fully adjusted model were 1.51 (95%CI: 1.37–1.66, $P < 2.00 \times 10^{-16}$) and 1.59 (95%CI: 1.44–1.76, $P < 2.00 \times 10^{-16}$) for ever-smoking and previous smoking, respectively (Table 1). There was no significant association observed between current smoking and the risk of UC (HR: 1.18, 95%CI: 1.00–1.39, $P = 0.051$). Table 2 shows that these positive associations were still significant in the stratified analysis (all $P < 0.001$). Albeit not significant, a trend towards a reverse association pattern between current smoking and UC risk was observed in those under 60 years (HR: 0.82, 95%CI: 0.61–1.10, $P = 0.179$). In addition, smoking cessation after 50 years old was associated with decreased risk of CD (HR: 0.12, 95%CI: 0.02–0.96, $P = 0.045$) and UC (HR: 0.25, 95%CI: 0.09–0.74, $P = 0.012$) in those with disease onset less than 60 years of age (Table 2).

We did not discover any significant association between genetic predisposition to smoking behaviors and the risk of CD and UC in the main MR analysis and sensitivity analyzes, and there was no horizontal pleiotropy of the used IVs (Supplementary Table 2). We identified seven outliers in the analysis of *SmkInit* and CD risk using the MR-PRESSO approach and the causal association became nominally significant after removing these outliers (beta=0.13, $P = 0.049$, Supplementary Table 3). However, no association survived FDR correction.

### Smoking-related DNA methylation and IBD risk

Among the 2,623 CpG sites that associate with smoking behaviors[14], 1,938 for CD and 1,939 for UC had available *cis*-mQTLs that could be used as proxies of smoking-related DNA methylation in the two-sample MR analyzes[15] (Supplementary Data 1). We discovered that altered methylation at 46 smoking-related CpG sites was significantly associated with the risk of CD (FDR corrected $P < 0.046$), including cg08540958 (located at *CCL2* gene body), cg17742416 (located at *DNMT3A* gene body), cg03599224 (located near *LTA/TNF* within the HLA class III locus), cg04641860 (located at *HNRNPF* 5'UTR), cg25607920, cg03884592, cg15937073, cg26038582, cg16145216 (all located at *HIVEP3* gene body), and cg21920570 (located at *MACROD1* gene body) (Fig. 1 and Supplementary Data 1). For UC, the altered

## Table 1 | Associations between smoking status and inflammatory bowel disease

| | Crohn's disease (cases, n = 1770) | | | | Ulcerative colitis (cases, n = 2889) | | | |
|---|---|---|---|---|---|---|---|---|
| | Crude model* | | Fully adjusted# | | Crude model* | | Fully adjusted# | |
| | HR (95%CI) | *P* | HR (95%CI) | *P* | HR | *P* | HR | *P* |
| Never smoking | ref | | ref | | ref | | ref | |
| Ever smoking | 1.51 (1.37, 1.65) | <2 × 10⁻¹⁶ | 1.49 (1.31, 1.69) | 5.83 × 10⁻¹⁰ | 1.45 (1.34, 1.56) | <2 × 10⁻¹⁶ | 1.51 (1.37, 1.66) | <2 × 10⁻¹⁶ |
| Previous smoking | 1.39 (1.25, 1.54) | 5.06 × 10⁻¹⁰ | 1.41 (1.23, 1.61) | 7.00 × 10⁻⁷ | 1.54 (1.42, 1.66) | <2 × 10⁻¹⁶ | 1.59 (1.44, 1.76) | <2 × 10⁻¹⁶ |
| Current smoking | 1.88 (1.64, 2.16) | <2 × 10⁻¹⁶ | 1.81 (1.50, 2.19) | 7.09 × 10⁻¹⁰ | 1.15 (1.01, 1.30) | 0.038 | 1.18 (1.00, 1.39) | 0.051 |
| Pack years | | | | | | | | |
| <= 20 | ref | | ref | | ref | | ref | |
| 20 – 40 | 1.05 (0.88, 1.25) | 0.623 | 1.10 (0.88, 1.38) | 0.393 | 1.12 (0.98, 1.29) | 0.107 | 1.09 (0.92, 1.29) | 0.313 |
| 40 – 60 | 1.22 (0.98, 1.51) | 0.070 | 1.34 (1.02, 1.75) | 0.036 | 1.08 (0.91, 1.29) | 0.396 | 1.08 (0.87, 1.33) | 0.485 |
| > 60 | 1.03 (0.81, 1.32) | 0.805 | 1.15 (0.85, 1.55) | 0.372 | 1.12 (0.93, 1.35) | 0.235 | 1.05 (0.83, 1.31) | 0.703 |
| Age of stopping smoking | | | | | | | | |
| <= 30 | ref | | ref | | ref | | ref | |
| 30 – 40 | 1.24 (0.98, 1.58) | 0.079 | 1.35 (1.00, 1.81) | 0.046 | 1.18 (0.99, 1.40) | 0.069 | 1.14 (0.93, 1.41) | 0.214 |
| 40 – 50 | 1.17 (0.90, 1.52) | 0.232 | 1.26 (0.90, 1.75) | 0.178 | 1.32 (1.10, 1.58) | 2.72 × 10⁻³ | 1.40 (1.12, 1.75) | 0.003 |
| > 50 | 1.89 (1.46, 2.43) | 9.49 × 10⁻⁷ | 1.79 (1.29, 2.50) | 0.001 | 1.50 (1.24, 1.81) | 2.59 × 10⁻⁵ | 1.48 (1.16, 1.87) | 0.001 |

*, the crude model was adjusted for age and sex. #, the fully adjusted model was adjusted age, sex, BMI, drinking status, physical activity, processed meat, body mass index, education level, and the first 10 genetic principal components. Cox regression model was used to evaluate the association between smoking status and IBD. All tests were two-sided and $P < 0.05$ was considered as significant. Source data are provided as a Source Data for Tables 1, 2 and Supplementary data7 (CD analysis) and Source Data for Tables 1, 2 and Supplementary Data 7 (UC analysis).
*HR* hazard ratio, *CI* confidence interval.

**Table 2 | Association between smoking status and IBD according to age stratification**

| | Crohn's disease (1770 cases) | | | | Ulcerative colitis (2889 cases) | | | |
|---|---|---|---|---|---|---|---|---|
| | Fully adjusted | | | | Fully adjusted | | | |
| | HR | LCI | UCI | P | HR | LCI | UCI | P |
| **<60 years old (CD: 576 cases; UC:797 cases)** | | | | | | | | |
| **Never smoking** | ref | ref | ref | ref | ref | ref | ref | ref |
| **Ever smoking** | 1.23 | 0.98 | 1.54 | 0.068 | 1.39 | 1.16 | 1.67 | $4.32 \times 10^{-4}$ |
| **Previous smoking** | 1.20 | 0.94 | 1.54 | 0.148 | 1.69 | 1.39 | 2.04 | $9.47 \times 10^{-8}$ |
| **Current smoking** | 1.29 | 0.95 | 1.74 | 0.103 | 0.82 | 0.61 | 1.10 | 0.179 |
| **Pack years of smoking** | | | | | | | | |
| **<= 20** | ref | ref | ref | ref | ref | ref | ref | ref |
| **20 – 40** | 0.84 | 0.56 | 1.26 | 0.403 | 0.73 | 0.52 | 1.02 | 0.065 |
| **40 – 60** | 0.79 | 0.45 | 1.38 | 0.405 | 0.82 | 0.53 | 1.29 | 0.397 |
| **> 60** | 0.85 | 0.48 | 1.49 | 0.571 | 0.75 | 0.47 | 1.21 | 0.237 |
| **Age of stopped smoking** | | | | | | | | |
| **<= 30** | ref | ref | ref | ref | ref | ref | ref | ref |
| **30 – 40** | 1.47 | 0.84 | 2.59 | 0.177 | 0.61 | 0.41 | 0.92 | 0.019 |
| **40 – 50** | 1.24 | 0.65 | 2.38 | 0.517 | 1.01 | 0.67 | 1.54 | 0.949 |
| **> 50** | 0.12 | 0.02 | 0.96 | 0.045 | 0.25 | 0.09 | 0.74 | $1.20 \times 10^{-2}$ |
| **>=60 years old (CD: 1194 cases; UC: 2092 cases)** | | | | | | | | |
| **Never smoking** | ref | ref | ref | ref | ref | ref | ref | ref |
| **Ever smoking** | 1.52 | 1.30 | 1.77 | $1.48 \times 10^{-7}$ | 1.46 | 1.30 | 1.63 | $6.28 \times 10^{-11}$ |
| **Previous smoking** | 1.45 | 1.23 | 1.71 | $6.58 \times 10^{-6}$ | 1.51 | 1.35 | 1.70 | $2.65 \times 10^{-12}$ |
| **Current smoking** | 1.83 | 1.43 | 2.34 | $1.25 \times 10^{-6}$ | 1.19 | 0.97 | 1.45 | 0.097 |
| **Pack years of smoking** | | | | | | | | |
| **<= 20** | ref | ref | ref | ref | ref | ref | ref | ref |
| **20 – 40** | 1.21 | 0.92 | 1.60 | 0.178 | 1.21 | 1.00 | 1.47 | 0.054 |
| **40 – 60** | 1.51 | 1.10 | 2.07 | 0.011 | 1.14 | 0.90 | 1.45 | 0.280 |
| **> 60** | 1.28 | 0.90 | 1.84 | 0.175 | 1.15 | 0.88 | 1.49 | 0.310 |
| **Age of stopped smoking** | | | | | | | | |
| **<= 30** | ref | ref | ref | ref | ref | ref | ref | ref |
| **30 – 40** | 1.12 | 0.79 | 1.58 | 0.540 | 1.32 | 1.02 | 1.69 | 0.033 |
| **40 – 50** | 1.01 | 0.69 | 1.49 | 0.957 | 1.38 | 1.06 | 1.80 | $1.80 \times 10^{-2}$ |
| **> 50** | 1.61 | 1.12 | 2.30 | 0.009 | 1.58 | 1.21 | 2.07 | $1.00 \times 10^{-3}$ |

The fully adjusted model was adjusted age, sex, BMI, drinking status, physical activity, processed meat, body mass index, education level, and the first 10 genetic principal components. All cases are incident cases. Cox regression was used to evaluate the association of smoking behavior with Crohn's disease and ulcerative colitis. All tests were two-sided and P < 0.05 was considered as significant. Source data are provided as Source Data for Tables 1, 2 and Supplementary Data 7 (CD analysis) and Source Data for Tables 1, 2 and Supplementary Data 7 (UC analysis).
*HR* hazard ratio, *LCI* lower 95%CI, *UCI* upper 95%CI.

methylation at 69 smoking-related CpG sites passed the FDR correction and was associated with UC risk (Fig. 2 and Supplementary Data 1), including cg27310486 (located at *CCDC71* gene body), cg17742416 (located at *DNMT3A* body), cg14647125 and cg23916869 (all located at *AHRR* gene body). Horizontal pleiotropy of the used instruments for cg20408402, cg27182159, cg07516196, and cg05232694 was detected by the MR-Egger approach (Supplementary Data 2).

### Replication of differentially methylated genetic signals in IBD
We additionally performed a genome-wide methylation analysis on CD and UC in the IBD-CHARACTER inception cohort to validate the association of altered DNA methylation at genetic loci related to smoking exposure. There were 343 IBD patients and 295 controls in the cohort, and 154 patients were classified as CD, 161 as UC, and 28 as unspecified IBD. There were 54 current and 103 former smokers in the IBD group. Smoking exposure details are displayed in Supplementary Table 4. The mean age of IBD patients was 34 [range 7-79] years, and 33 [range 3-79] years in the control[11], respectively (Supplementary Table 4). The differentially methylated CpG sites and their related loci are displayed in Supplementary Data 3 and 4; and the results of sensitivity analysis are presented in Supplementary Data 5 and 6. When compared with the epigenetic MR analysis, the association between altered DNA methylation at five genetic loci (regulated by nine smoking-related DNA methylation CpG sites) and CD risk was validated, including *LTA/TNF*, *DNMT3A*, *HIVEP3*, *HNRNPF*, and *MACROD1* (Tables 3 and 4). In particular, altered methylation at two CpG sites located in different regions of the *LTA* locus, i.e., cg03599224 (located near *LTA/TNF* within the HLA class III locus) and cg16280132 (located at 5'UTR of *LTA* gene) had a consistent effect on IBD, judged by both their potential effect on gene expression and the beta value derived in epigenetic MR and IBD EWAS (Tables 3 and 4). The findings for UC risk, including altered DNA methylation at cg14647125 and cg23916896 (all located at the *AHRR* gene body), and the association of aberrant DNA methylation at *DNMT3A* loci, were also successfully replicated (Tables 3 and 4).

### Analysis of shared variants involved in determining specific CpG site methylation and IBD susceptibility
We found strong co-localization evidence for one of the nine CpG sites that exhibited significant association with CD risk. As displayed in Supplementary Table 5 and Fig. 3, methylation at cg03599224 (located near *LTA/TNF* within the HLA class III locus) had a 100% posterior probability of sharing a causal variant with CD susceptibility. Since there were fewer than 10 available mQTLs for cg17742416 (located at *DNMT3A*), this CpG was excluded from the colocalization analysis. For UC, none of the CpG sites was discovered to have strong co-localization evidence (Supplementary Table 5). We did not identify any mQTL that interacted significantly with all smoking phenotypes after FDR correction (Supplementary Data 7). Therefore, we did not conduct any further stratification analysis.

### Discussion
In this study, we first confirmed in the extended population studied in the UK Biobank that current smoking and previous smoking are correlated with elevated CD and UC risks, respectively. Interestingly, smoking cessation after the age of 50 was found to be linked with decreased risks of the two disease subtypes in those presenting within 10 years of cessation. DNA methylation alteration at cg17742416 [*DNMT3A*] was linked with the risk of both CD and UC, while cg03599224 (located near *LTA/TNF* within the *HLA* class III locus) was associated with CD risk, and cg14647125 and cg23916896 [all located in *AHRR*] were linked to UC risk.

Established epidemiological data strongly suggest the substantial impact of the exposome on the pathogenesis of IBD[16]. Of all environmental influences, smoking has been the most intensely investigated, and several studies have linked cigarette smoking to the development and course of IBD[5,16]. In CD, studies demonstrated a significant association between current smoking and an elevated risk of disease development, postoperative recurrence, and a more aggressive disease course[17]. In contrast, former smoking and current smoking have been associated with higher UC risk and lower relapse rate compared to non-smokers, respectively[18]. Previous MR studies reported inconsistent evidence on the causality of smoking traits with IBD risk[19–21], and highlighted the potential of inadequate strength of genetic IVs used for smoking. We have previously suggested that DNA methylation markers have the potential to explain a higher proportion of the variance in

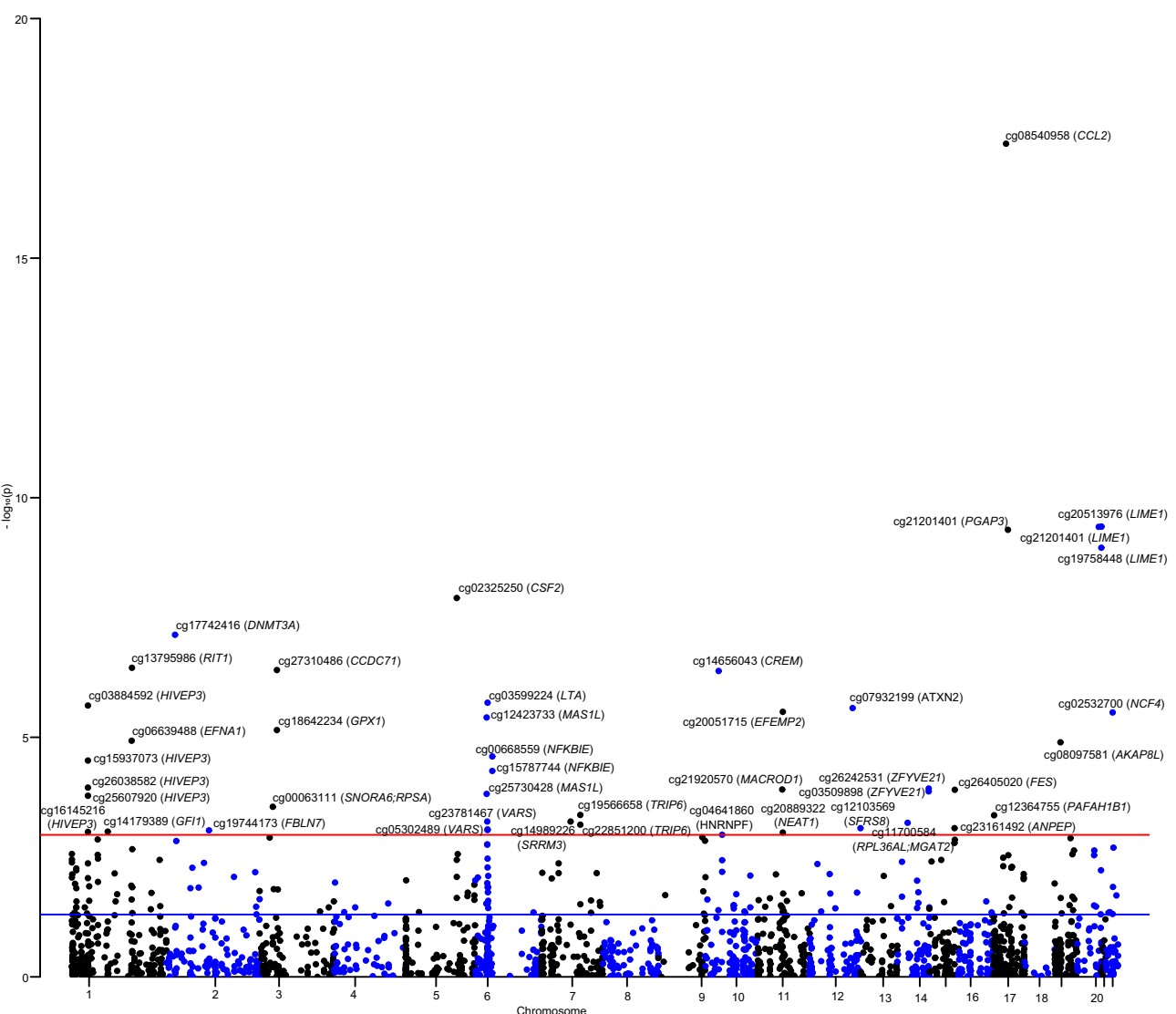

**Fig. 1 | The association of gene regulated by smoking-related DNA methylation with the risk of Crohn's disease.** This analysis was conducted basing on the data from the smoking EWAS (Joehanes R, et al., N = 15,907) and GWAS summary level data of IBD (de Lange KM, et al., N = 40,266, 12,366 UC cases). MR estimates were measured through the Wald ratio approach and combined using the IVW method if there were more than one independent instrumental variant. All tests were two-sided and adjusted for multiple comparisons. Manhattan plot showing the -log₁₀(*p*) of association between each CpG site with the risk of CD on the y-axis against genomic position on the x-axis. Gene names are italicized. The blue line means *P* = 0.05, and the red line means the threshold of false discovery rate correction. Source data are provided as Source Data for Figs. 1 and 2.

smoking exposure and serve as better proxies than polygenic scores of the exposure (60.9% *vs* 2.8%)[22]. In this study, we used genetic instruments of DNA methylation markers related to smoking and performed MR analysis using the largest IBD GWAS to enhance the study power. Our study with over half a million participants corroborated these findings on IBD risk and also provided evidence supporting the involvement of tobacco smoking in IBD pathogenesis through epigenetics.

In line with our findings, an umbrella review of meta-analyzes indicated a consistent relationship between current smoking habits and the susceptibility to CD (OR: 1.76; 95% CI: 1.40-2.22)[23]. They reported that current smoking is negatively correlated with the risk of developing UC (OR:0.58, 95%CI: 0.45-0.75); our study also found a similar trend in the population under 60, albeit not statistically significant. Furthermore, studies have indicated the beneficial and harmful effects of smoking cessation on the occurrence, prognosis, and treatment of CD and UC[4,24,25], respectively. Of interest, we observed a benefit of smoking cessation after 50 years of age on both CD and UC risk in those under 60 years but not in those older than 60. This protective effect may be due to the fact that smoking cessation might allow for recovery of the DNA methylation levels of smokers[26,27]. Other immunological mechanisms associated with ageing may also be relevant in the observed age-related changes, including differential effects on memory B cells[28]. Of particular note is the fact that the effect of smoking cessation was seen both in CD as well as UC, and this is apparently at odds with the well-evidenced and accepted association of smoking with susceptibility to IBD in young adults[29,30]. The latter is of interest in the light of emerging data that the phenotypic presentation, as well as complication profile and response to therapy, seems to differ between early-onset and late-onset IBD[31–33]. The current observations provide further evidence that there may be age-related heterogeneity in disease pathogenesis. Thus, aetiological factors involved in late-onset disease may differ from those in disease with onset in childhood or early-adulthood, a fact which may be important in designing prevention strategies.

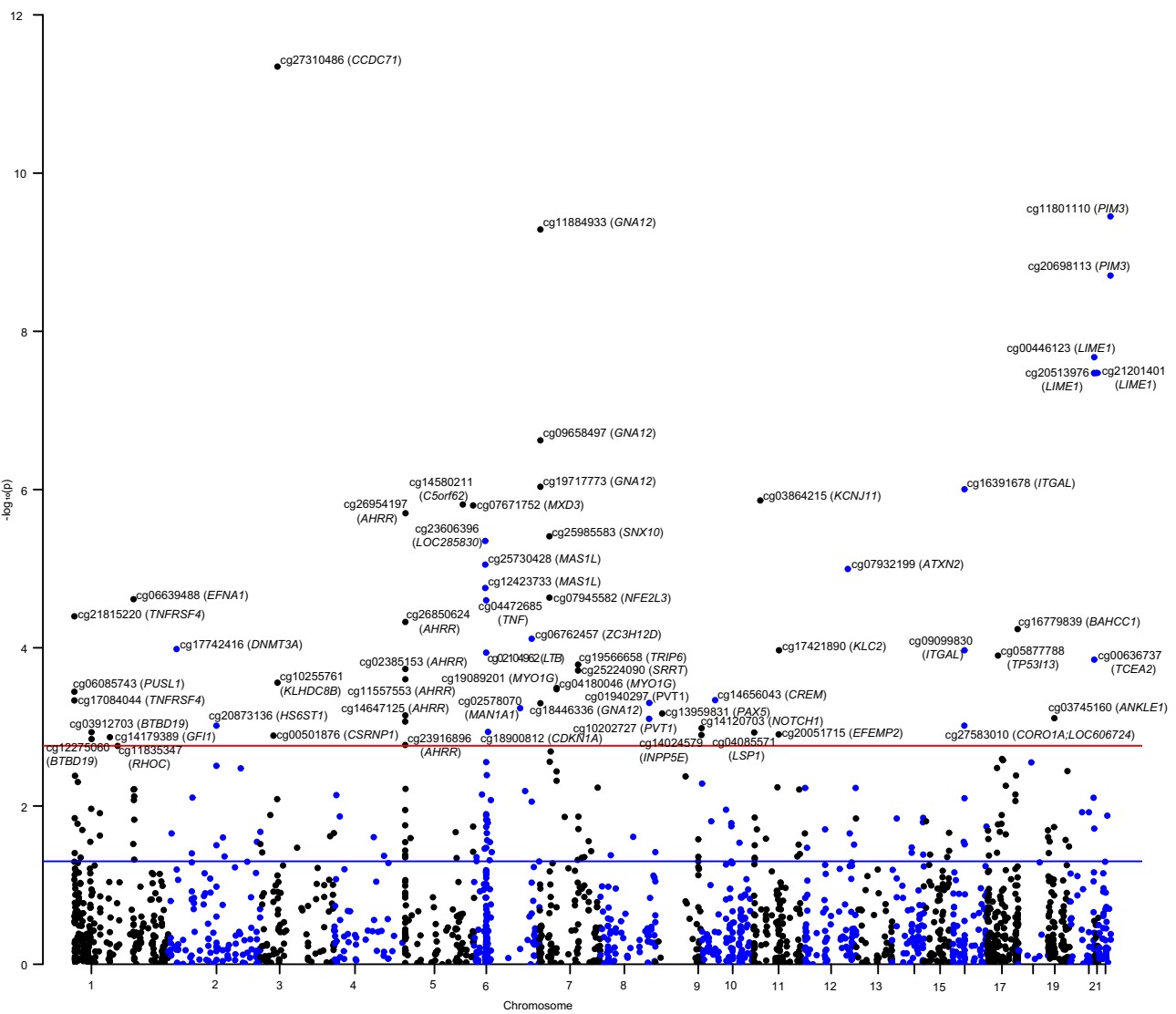

**Fig. 2 | The association of gene regulated by smoking-related DNA methylation with the risk of ulcerative colitis.** This analysis was conducted basing on the data from the smoking EWAS (Joehanes R, et al., $N = 15,907$) and GWAS summary level data of IBD (de Lange KM, et al., $N = 45,975$, 12,194 CD cases). MR estimates were measured through the Wald ratio approach and combined using the IVW method if there were more than one independent instrumental variant. All tests were two-sided and adjusted for multiple comparisons. Manhattan plot showing the $-\log_{10}(p)$ of association between each CpG site with the risk of CD on the y-axis against genomic position on the x-axis. Gene names are italicized. The blue line means $P = 0.05$, and the red line means the $p$-value threshold of FDR correction. Source data are provided as Source Data for Figs. 1 and 2.

To date, several immune-mediated pathways have been linked to smoking in IBD pathogenesis, but there is no consensus regarding either the active component of tobacco or the pathways involved[5]. Some of the mechanisms studied include the role of oxidative stress, epithelial barrier integrity, immune responses, alterations in the microbiome, and pro-inflammatory epigenetic modifications[34,35]. There is growing evidence of a differential methylome in IBD compared to controls, replicated across several populations in the UK, Europe, and North America[12,36]. Furthermore, it is well-established that smoking can influence DNA methylation patterns[8,9]. However, the biological pathway that could explain the differential effects of smoking on UC and CD remains unclear. Hypothesizing that the epigenome is uniquely positioned to bridge environmental exposures and genetic susceptibility, our study investigated the causality between cigarette smoking, DNA methylation, and the risk of IBD, by applying MR analysis. Epigenetic MR analysis identified 46 smoking-related loci with differential methylation in CD, and 69 in UC. Amongst these, several key genes were validated as having strong evidence in the next stage of our experiments. Among them, cg17742416 within *DNMT3A* was the only CpG associated with both diseases.

DNA methyltransferases (DNMTs) represent a family of enzymes that establish and maintain DNA methylation patterns across the genome. GWAS studies have indicated an association between polymorphisms in the *DNMT3A* locus and an increased risk of CD[37]. *DNMT3A* has emerged as a pivotal regulator in orchestrating intestinal epithelial homeostasis and modulating the response to tissue damage, thereby implicating impaired *DNMT3A* function in the pathogenesis of IBD[38]. Mechanistically a loss of functional *DNMT3A* in intraepithelial cells in *Dnmt3a*$^{ΔIEC}$ mice[38], was shown to increase susceptibility to experimental colitis. Moreover, *DNMT3A*-mediated epigenetic modulation of gene expression was indispensable for the establishment of precise epithelial junctional zones, preservation of intestinal homeostasis, and maintenance of optimal epithelial barrier function[38]. This is especially relevant in CD since altered intestinal permeability is associated with relapse and

**Table 3 | Differentially methylated genetic loci identified from the epigenetic MR of smoking in relation to the risk of Crohn's disease and ulcerative colitis**

| Outcome | Gene | | Epigenetic MR | | | | |
|---|---|---|---|---|---|---|---|
| | | CpG site | Region | beta | se | *P*-value | FDR |
| **Crohn's disease** | *LTA* | cg03599224 | body | −0.493 | 0.104 | $1.91 \times 10^{-6}$ | $3.08 \times 10^{-4}$ |
| | *DNMT3A* | cg17742416 | body | 0.417 | 0.077 | $7.30 \times 10^{-8}$ | $1.77 \times 10^{-4}$ |
| | *HIVEP3* | cg25607920 | body | 0.141 | 0.038 | $1.67 \times 10^{-4}$ | $1.08 \times 10^{-2}$ |
| | | cg03884592 | body | 0.142 | 0.030 | $2.17 \times 10^{-6}$ | $3.23 \times 10^{-4}$ |
| | | cg15937073 | body | 0.130 | 0.031 | $3.07 \times 10^{-5}$ | $2.70 \times 10^{-3}$ |
| | | cg26038582 | body | 0.131 | 0.034 | $1.12 \times 10^{-4}$ | $9.00 \times 10^{-3}$ |
| | | cg16145216 | body | 0.131 | 0.040 | $9.42 \times 10^{-4}$ | $4.14 \times 10^{-2}$ |
| | *HNRNPF* | cg04641860 | 5'UTR | −0.319 | 0.098 | $1.09 \times 10^{-3}$ | $4.59 \times 10^{-2}$ |
| | *MACROD1* | cg21920570 | body | 0.206 | 0.054 | $1.23 \times 10^{-4}$ | $9.00 \times 10^{-3}$ |
| **Ulcerative colitis** | *AHRR* | cg14647125 | body | −0.298 | 0.088 | $7.16 \times 10^{-4}$ | $2.62 \times 10^{-2}$ |
| | | cg26850624 | body | 0.105 | 0.026 | $4.70 \times 10^{-5}$ | $3.25 \times 10^{-3}$ |
| | | cg26954197 | 3'UTR | 0.355 | 0.075 | $2.00 \times 10^{-6}$ | $2.28 \times 10^{-4}$ |
| | | cg17924476 | body | 0.184 | 0.055 | $8.58 \times 10^{-4}$ | $2.97 \times 10^{-2}$ |
| | | cg23916896 | body | −0.241 | 0.077 | $1.69 \times 10^{-3}$ | $4.81 \times 10^{-2}$ |
| | *DNMT3A* | cg17742416 | body | 0.295 | 0.076 | $1.04 \times 10^{-4}$ | $6.00 \times 10^{-3}$ |

Gene names are italicized. MR estimates were measured through the Wald ratio approach and combined using the IVW method if there were more than one independent instrumental variant. All tests were two-sided and adjusted for multiple comparisons.
*MR* Mendelian randomization study, *5'UTR* 5' untranslated region, *3'UTR* 3' untranslated region, *Body* Gene body, *FDR* false discovery rate.

disease course, and increased permeability markers have been shown to precede the onset of CD by several months[39]. In UC, in vitro and in vivo experiments suggest that miR-182-5p can cause a disease exacerbation by inhibiting the *Wnt*/β-catenin signaling pathway through *DNMT3A*-mediated methylation of SMARCA5[40]. In addition, *DNMT3A* induces intestinal epithelial cell barrier dysfunction through TNFSF13-mediated interactions between intestinal epithelial cells and B cells that can lead to UC progression[41]. It has also been found that smoking-induced hypermethylation of *DNMT3A* and subsequent gene dysfunction could cause some of the changes associated with *DNMT3A* deficiency in myelopoiesis. Inborn *DNMT3A* hemizygosity (Tatton-Brown-Rahman syndrome) results in leukocyte features with increased neutrophil-to-lymphocyte ratio, reduced B-cell progenitors, and elevated IL-6[42]. *DNMT3A* mutations are also known for causing clonal hematopoiesis (and acute myeloid leukemia), which is more common in UC, and which associates with greater production of interferon-gamma[43]. Collectively, these findings implicate DNMT3 both in maintenance of gut integrity as well as in immune homeostasis and provide a further impetus to investigate in detail the contribution of *DNMT3A* function in IBD pathogenesis.

Aryl-hydrocarbon receptor repressor (*AHRR*) and several CpG probes within this gene have been strongly linked to smoking exposure. Of note, alteration in cg05575921 methylation has been demonstrated to have very high accuracy in differentiating between smokers and non-smokers in whole blood as well as in other tissue/cell types[44,45]. Our study implicates this CpG probe and finds five other CpG sites located in the *AHRR* gene in smoking-related differential DNA methylation. Two of them (cg14647125 and cg23916896) showed the same effect direction on UC risk with the findings from genome-wide DNA methylation analysis of UC. The Aryl-hydrocarbon receptor (AHR) influences immune cell activation and homeostasis in the intestine and is usually highly expressed in barrier tissues such as the gut[46], acting as a sensor for environmental toxins and dietary compounds[46,47]. In fact, AhR signaling has been demonstrated to play key roles in regulating Th17/Treg balance and the development and function of innate lymphoid cells[48,49]. Reduced representation bisulfite sequencing (RRBS) has

identified that the most significant differentially methylated region occurred at a poised enhancer in the *AHRR*, which may upregulate *AHRR* mRNA in a monocyte-specific way by activating the AHRR enhancer[50]. In addition, AHR[-/-] mice exposed to dextran sulfate-sodium-induced colitis demonstrated a worse phenotype compared to the controls[51]. Future studies exploring the complex relationship between smoking, epigenetic and functional alteration of *AHRR* and IBD disease onset and progression will be of great interest.

Our study also showed that methylation at cg03599224 (located near *LTA/TNF* within the *HLA* class III locus) was linked to CD risk. Whilst this may affect transcription across this region, given the extent of linkage disequilibrium, effects on either lymphotoxin itself, or *TNF* transcription are relevant and highly plausible in the setting of IBD. TNF-α and LTA (TNF-β) are structurally related cytokines belonging to the TNF superfamily, with their genes closely situated within the *HLA* class III locus. The *TNF* position-308 and *LTA NcoI* polymorphisms exhibit robust linkage disequilibrium and are proposed to affect *TNF* transcription and secretion[52,53]. Interestingly, a recent study showed that *TNF* mRNA and protein levels in human monocytic THP-1 cells were significantly increased when exposed to aqueous smoke extracts of conventional cigarette or alternative smoking products[54]. Short-term cigarette extract (CSE) stimulation affects the maturation status of dendritic cells (DCs) represented by the elevated production of TNF-α, IL-12, and IL-6, upregulated expression of MHCII, CD83, CD86, and CD40, and reduced antigen uptake capacity, while long-term CSE exposure suppresses the functional DCs development as revealed by a decreased cytokine secretion and impaired ability to stimulate T lymphocytes[55]. The *LTA* 252 A > G polymorphism was also reported to regulate the inflammatory effects of smoking and oxidative stress[56]. Lymphotoxin-α/TNF-β (*LTA*) plays a role in atherosclerotic and may also lead adhesion molecules and cytokines in vascular endothelial cells and smooth muscle cells[57]. *LTA* formed granulomas in leprosy in response to *M. leprae* infection, and might also affect autoimmune disease (including CD) similarly[58]. Genetic polymorphisms of *LTA* have also been found to impact the effect of anti-TNF-α therapies on CD[59,60]. Given the role of anti-TNF-directed

**Table 4 | Validation of differentially methylated genetic loci in the genome-wide methylation analysis of Crohn's disease and ulcerative colitis**

| Outcome | Gene | Genome-wide methylation analysis | | | | | |
|---|---|---|---|---|---|---|---|
| | | CpG site | Region | beta | se | *P*-value | FDR |
| **Crohn's disease** | *LTA* | cg16280132 | 5'UTR | 0.009 | 0.004 | $3.43 \times 10^{-2}$ | $4.49 \times 10^{-2}$ |
| | *DNMT3A* | cg23393100 | body | 0.002 | 0.001 | $1.46 \times 10^{-2}$ | $3.40 \times 10^{-2}$ |
| | | cg17207266 | body | 0.008 | 0.002 | $4.36 \times 10^{-5}$ | $1.42 \times 10^{-3}$ |
| | | cg05544807 | body | 0.012 | 0.003 | $2.77 \times 10^{-4}$ | $3.61 \times 10^{-3}$ |
| | | cg20702417 | body | 0.010 | 0.003 | $7.66 \times 10^{-4}$ | $7.11 \times 10^{-3}$ |
| | | cg03314052 | body | 0.008 | 0.003 | $1.69 \times 10^{-2}$ | $3.43 \times 10^{-2}$ |
| | | cg11779362 | body | 0.002 | 0.001 | $1.03 \times 10^{-2}$ | $3.04 \times 10^{-2}$ |
| | *HIVEP3* | cg00970057 | body | 0.009 | 0.003 | $3.59 \times 10^{-3}$ | $1.56 \times 10^{-2}$ |
| | | cg09501963 | body | 0.005 | 0.002 | $2.83 \times 10^{-2}$ | $4.34 \times 10^{-2}$ |
| | | cg24051040 | body | 0.004 | 0.002 | $3.87 \times 10^{-2}$ | $4.49 \times 10^{-2}$ |
| | *HNRNPF* | cg00326958 | 5'UTR | −0.007 | 0.003 | $8.06 \times 10^{-3}$ | $3.00 \times 10^{-2}$ |
| | *MACROD1* | cg10612274 | body | 0.025 | 0.007 | $2.06 \times 10^{-4}$ | $3.35 \times 10^{-3}$ |
| | | cg17590162 | body | 0.013 | 0.005 | $1.30 \times 10^{-2}$ | $3.20 \times 10^{-2}$ |
| | | cg12424921 | body | 0.010 | 0.005 | $2.84 \times 10^{-2}$ | $4.34 \times 10^{-2}$ |
| | | cg04582871 | body | 0.011 | 0.005 | $3.77 \times 10^{-2}$ | $4.49 \times 10^{-2}$ |
| | | cg09908110 | body | 0.001 | 0.001 | $4.56 \times 10^{-2}$ | $4.69 \times 10^{-2}$ |
| | | cg09375205 | body | 0.006 | 0.002 | $1.56 \times 10^{-2}$ | $3.43 \times 10^{-2}$ |
| | | cg02374207 | body | 0.002 | 0.001 | $1.12 \times 10^{-2}$ | $3.12 \times 10^{-2}$ |
| **Ulcerative colitis** | *AHRR* | cg22356527 | body | −0.005 | 0.002 | $3.04 \times 10^{-2}$ | $4.47 \times 10^{-2}$ |
| | | cg19602050 | body | −0.001 | 0.000 | $1.72 \times 10^{-2}$ | $3.59 \times 10^{-2}$ |
| | | cg12806681 | body | 0.006 | 0.002 | $5.53 \times 10^{-3}$ | $2.71 \times 10^{-2}$ |
| | | cg02088390 | body | −0.002 | 0.001 | $2.33 \times 10^{-2}$ | $4.37 \times 10^{-2}$ |
| | | cg26703534 | body | 0.013 | 0.004 | $1.22 \times 10^{-3}$ | $1.25 \times 10^{-2}$ |
| | | cg11894422 | body | 0.005 | 0.002 | $3.21 \times 10^{-2}$ | $4.47 \times 10^{-2}$ |
| | | cg21161138 | body | 0.013 | 0.004 | $3.52 \times 10^{-3}$ | $2.16 \times 10^{-2}$ |
| | | cg23576855 | body | 0.034 | 0.016 | $3.71 \times 10^{-2}$ | $4.47 \times 10^{-2}$ |
| | | cg11557553 | body | −0.004 | 0.002 | $4.66 \times 10^{-2}$ | $4.87 \times 10^{-2}$ |
| | | cg25648203 | body | 0.010 | 0.004 | $1.51 \times 10^{-2}$ | $3.42 \times 10^{-2}$ |
| | *DNMT3A* | cg26544247 | body | 0.006 | 0.003 | $3.71 \times 10^{-2}$ | $4.47 \times 10^{-2}$ |
| | | cg20702417 | body | 0.006 | 0.003 | $4.85 \times 10^{-2}$ | $4.93 \times 10^{-2}$ |
| | | cg15843262 | body | 0.005 | 0.002 | $5.10 \times 10^{-3}$ | $2.65 \times 10^{-2}$ |
| | | cg17207266 | body | 0.006 | 0.002 | $4.35 \times 10^{-3}$ | $2.45 \times 10^{-2}$ |
| | | cg27369452 | body | 0.001 | 0.001 | $8.24 \times 10^{-3}$ | $3.05 \times 10^{-2}$ |
| | | cg05544807 | body | 0.009 | 0.003 | $2.81 \times 10^{-3}$ | $2.00 \times 10^{-2}$ |
| | | cg11779362 | body | 0.002 | 0.001 | $4.91 \times 10^{-2}$ | $4.93 \times 10^{-2}$ |

Gene names are italicized. These loci were validated by a genome-wide DNA methylation analysis of inflammatory bowel disease[11]. All tests were two-sided and adjusted for multiple comparisons.
*5'UTR* 5′ untranslated region, *Body* Gene body, *FDR* false discovery rate.

therapies in both CD and UC, this finding provides an important avenue for mechanistic work.

Our study has several strengths in study design and analysis. This is one of the few studies which investigates and shows the influence of smoking-related DNA methylation on IBD risk, and as such provides potentially critical insights into the pathogenesis and the pathways involved in gene-environmental interactions, DNA methylation modulation, and immune dysfunction. We build on previous datasets that we and other investigators have generated, and we used well-established and validated MR techniques to investigate the causal association between smoking behavior, smoking-related blood DNA methylation and IBD risk, which strengthens the reliability of causal inference by diminishing residual confounding and reverse causality. Our study mainly focused on the European population, which has allowed us to minimize any effect of confounding caused by population structure. However, in turn, this aspect also restricts the generalizability of these findings to other ethnic populations. Once GWAS and EWAS data on other ethnicities become available, these findings will be explored.

In our statistical methodology, we acknowledge as a potential limitation (common to MR studies) the fact that using SNPs as IVs to infer causality in MR design may suffer from horizontal pleiotropy (i.e., IVs might be associated with unmeasured confounders[61]), especially for DNA methylation changes proxied by a few SNPs. A more liberal significance threshold was used in the genome-wide methylation profiling, which is based on a prior hypothesis regarding the loci involved, rather than a threshold associated with hypothesis-free analysis for those genetic loci identified in the epigenetic MR analysis. Furthermore, it is important to note that all findings remained significant when FDR correction was conducted for those validated signals (65 for CD and 135 for UC).

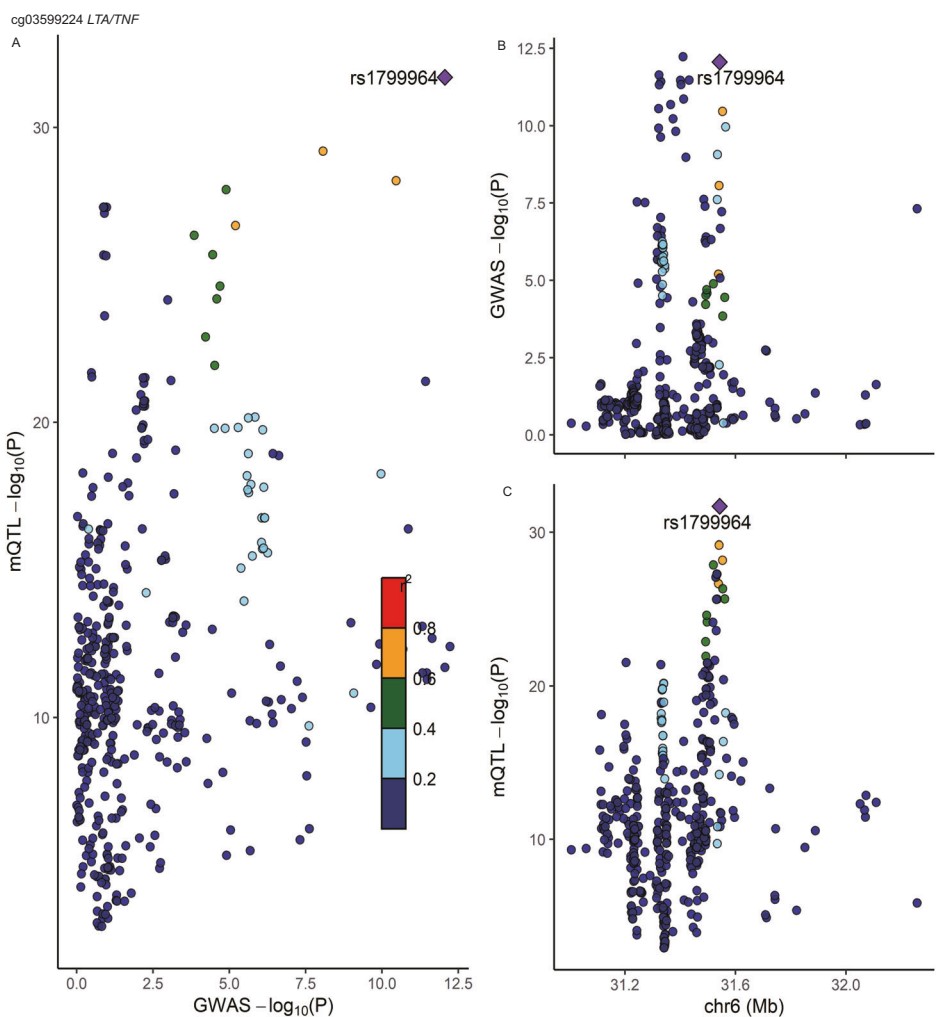

**Fig. 3 | Regional plot of colocalization evidence of CpG site methylation and Crohn's disease susceptibility.** Each dot represents a SNP. Gene names are italicized. Panel A displays the -log$_{10}$($p$) of the SNP and mQTL in corresponding GWAS and EWAS. In the Panel B and C, the x-axis demonstrates the base position of the SNP and mQTL, and the y-axis is the -log$_{10}$($p$) of the SNP in genome-wide association study and mQTL in genome-wide DNA methylation analysis, respectively. SNP that passed a colocalization threshold of posterior of H4 > 0.8 are highlighted in purple. Source data are provided as Source Data for Fig. 3.

In addition to statistical and analytical considerations, there are also biological issues to consider when interpreting these data. Changes in methylation are cell type specific, and immune cell subtypes may potentially alter over time due to smoking, or disease activity, or drug therapy; and this may affect the observed smoking-related changes in DNA methylation and transcription[62]. Since the smoking-related DNA methylation in CpG sites was obtained from a cross-sectional EWAS, we were unable to evaluate the time course, kinetics and dynamics of cigarette smoking on DNA methylation. Further longitudinal studies to explore the causal association between smoking and DNA methylation over time are warranted. It is reported that methylation at some CpG sites reverts to some extent after quit smoking, whereas certain CpG sites remain differentially methylated even after quit smoking for 35 years[26]. In this context, it is possible that the initial identification of mQTLs may have been affected by the characteristics of the study population, and we may underestimate the importance of smoking. As there were no sex- and age-stratified summary-level data available for CD and UC, we were unable to conduct this stratified MR analysis. These MR analyzes could be added when specific summary-level data would become available in the future.

It is also important to note that we have focused on late-onset IBD in this study, given the ages of patients recruited into the UK Biobank. Whether these findings are generalizable to patients with disease onset in earlier life remains an important issue to address. As discussed above, immunological and/or epigenetic changes associated with ageing preclude extrapolating our data to disease with onset in childhood, or early adult life when incidence is highest. Finally, whilst we demonstrated robust and validated data, we have concentrated on epigenetic signals in the blood and the EWAS of smoking-related DNA methylation was conducted by using blood samples. It should be considered that the DNA methylation signature is diverse in different tissues and immune cell subtypes[28], further research using data derived from intestinal tissues and different immune cell subtypes is warranted.

In conclusion, this study offers important insights into the potentially causative involvement of smoking in the pathogenesis of IBD. We confirm the unequivocal association of smoking with disease risk in the largest cohort reported to date. Moreover, we provide detailed evidence that epigenetic alterations are key in this relationship. Our data suggest that epigenetic dysregulation of *DNMT3A*, *LTA/ TNF*, and *AHRR* may mediate the pathogenic effects of smoking on the pathogenesis of IBD, thereby providing insight into functional and mechanistic studies in the immune system in the gut as well as in the circulation to explore the potential pathways for the disease. Therefore, these findings provide potential avenues for the exploration of disease pathogenesis as well as for the development of new

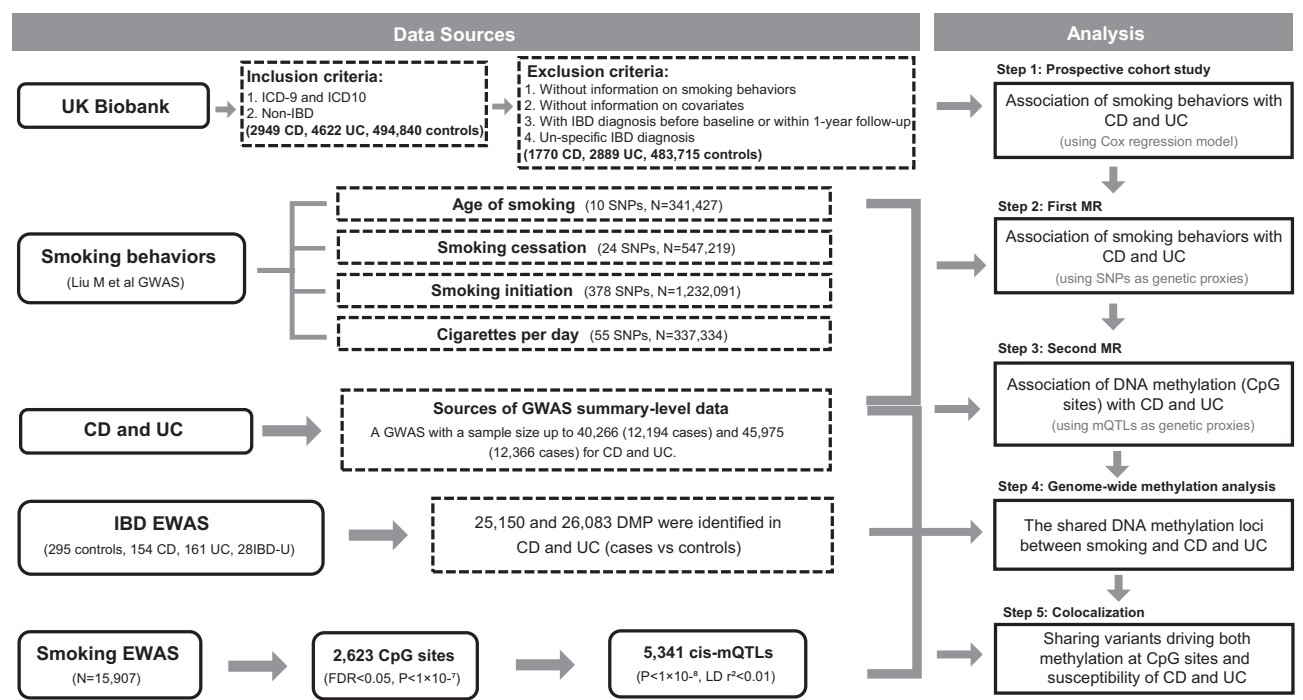

**Fig. 4 | The study design of this study.** CD, Crohn's disease. UC, ulcerative colitis. SNPs, single nucleotide polymorphisms. MR, Mendelian Randomization. GWAS, genome-wide association study. mQTL, methylation quantitative trait loci. IBD, inflammatory bowel disease. IBD-U, unspecific inflammatory bowel disease. DMP, differentially methylated position. EWAS, epigenome-wide association study.

pharmacological and non-pharmacological interventions based on epigenetic mechanisms to prevent and treat IBD.

## Methods

The study design is displayed in Fig. 4. We aimed to investigate the biological pathway whereby smoking may be implicated in the development of IBD. Sex differences were not the primary focus of this study; hence, our main emphasis was directed towards the overall population. In the cohort study and the genome-wide DNA methylation study, information about gender/sex were obtained and taken as a covariate in the analysis. In the UK Biobank, gender information was obtained through questionnaires. In the IBD-CHARACTER inception cohort, information about sex was collected by questionnaires and validated through technical analysis based on the median intensities of sex chromosome probes. Since MR analyzes were based on the summary statistics of IBD, we could not obtain the data stratified by sex, but the original GWAS has validated sex information through technical analysis and adjusted for this covariate. All participants who participated in the study provided informed content. Since this study is a secondary analysis based on previously published data, information on participant compensation was not applicable. The UK Biobank obtained ethical approval from the North West-Haydock Research Ethics Committee (REC reference: 16/NW/0274). The genome-wide methylation association study, conducted on a subset of the IBD-CHARACTER inception cohort, was approved by the Regional Committee for Medical Research Ethics, South-Eastern Norway (REK sør-øst 2009/2015) and received endorsement from the privacy protection representative at Akershus University Hospital (13-123). The ethical approval of published GWASs and EWASs was obtained from their corresponding review board.

### The population-based prospective cohort study
**Study population.** The UK Biobank comprises approximately 500,000 individuals aged between 40 and 69 years who were recruited from a total of 22 assessment centers located across the United Kingdom

during the period spanning from 2006 to 2010[63]. It provides a wide range of health-related phenotypic and genotypic information for each individual, including demographic data, lifestyle factors, physical examinations, biological samples, and genetic profiling[64].

**Smoking behaviors and covariates.** Three smoking behavior phenotypes, including current smoking status, cumulative pack years of smoking, and age at cessation of smoking, were examined in the observational study. Additionally, covariates such as gender, age, alcohol consumption status, physical activity level, processed meat intake, body mass index (BMI), educational attainment level, and the first 10 genetic principal components were taken into account. The information on these factors was acquired from questionnaires at baseline and follow-up, physical measurements, and health registries. Smoking status comprises never smoking, ever smoking, previous smoking, and current smoking. The variables of pack years of smoking and age at cessation were transformed into categorical factors.

**Ascertainment of CD and UC.** CD and UC cases with comprehensive data on smoking behaviors and covariates were identified through the electronic record of the UK Biobank. Complete follow-up was tracked to February 20, 2022. CD and UC cases were ascertained by using the 9th and 10th International Classification of Disease codes (Supplementary Table 6). Briefly, Participants without complete information on smoking behaviors, covariates, and genetic information, or with IBD diagnosis before baseline, or un-specific IBD diagnosis were excluded from the dataset. With a mean follow-up of 11 years, a total of 481,743 (1770 incident cases) and 482,862 (2889 incident cases) individuals were finally included in the analyzes for CD and UC, respectively.

**Statistical analysis.** Cox regression models were applied to evaluate the associations between smoking behaviors and the risks of newly-incident CD and UC. The crude model was adjusted for age and sex, whereas the fully adjusted model further accounted for alcohol consumption, physical activity level, intake of processed meat, BMI,

education level, and the first 10 genetic principal components. To assess the association between smoking behaviors and IBD risk for early- and late-onset IBD, we additionally performed stratification analysis in two age groups (< 60-year-old and >= 60-year-old)[65]. All statistical tests were conducted using a two-sided approach, and a significance level of $P < 0.05$ was employed. The analyzes were performed utilizing the R software (version 4.1.3).

### Two-sample mendelian randomization study

**Smoking behaviors and IBD risk**. Genetic instruments of smoking behaviors were obtained from a genome-wide association study (GWAS) conducted among 1.2 million European participants[66]. Researchers have identified 10 SNPs significantly associated ($P < 5 \times 10^{-8}$) with a continuous smoking initiation phenotype, namely the age of initiation of regular smoking (*AgeSmk*), and 378 SNPs significantly associated with a binary smoking initiation phenotype, specifically whether an individual has ever smoked regularly (*SmkInit*)[66] The heaviness of smoking was assessed by quantifying the average daily cigarette consumption (*CigDay*) among smokers and 55 instrumental variants (IVs) were discovered at genome-wide significance. In addition, 24 genetic variants representing smoking cessation (*SmkCes*) were identified at genome-wide significance by comparing current versus former smokers.[66]SNPs in linkage disequilibrium (LD, $r^2 > 0.01$) were excluded by performing SNP-clumping. Nine, 314, 40, and 18 SNPs were employed as IVs for four smoking phenotypes (i.e., *AgeSmk*, *SmkInit*, *CigDay*, and *SmkCes*). Trait variance explained by used IVs for *AgeSmk*, *SmkInit*, *CigDay*, and *SmkCes* was 0.11%, 4.69%, 1.07%, and 0.81%, respectively. To discover weakness instruments, F-statistics were calculated for these genetic instruments, and $F_{\text{-statistic}} < 10$ was considered to be weakness instruments. Information on these IVs is displayed in Supplementary Data 8. Genetic associations with CD and UC were derived from summary statistics of a large GWAS conducted in European populations, including 40,266 (12,194 cases) and 45,975 (12,366 cases) participants for CD and UC, respectively[15]. To appraise the potential causal associations between genetic predisposition to smoking behaviors and the risks of CD and UC, four phenotypes (i.e., *AgeSmk*, *SmkInit*, *CigDay*, and *SmkCes*) were treated as exposures. We adopted the inverse variance weighted (IVW) MR method as the main analysis, and the MR-Egger, weighted median, simple mode, weighted mode, and the MR pleiotropy residual sum and outlier (MR-PRESSO) approaches as sensitivity analyzes.

**Smoking-related alterations in blood DNA methylation and IBD risk.** The association estimates between smoking and DNA methylation were derived from a meta-analysis of genome-wide DNA methylation, which was conducted among 15,907 individuals[27]. DNA extracted from whole blood, CD4 + T cells, or monocytes was utilized to measure methylation. Bisulfite conversion of DNA was performed using the Zymo EZ DNA methylation kit, and the Infinium HumanMethylation 450 BeadChip, which contains 485,512 CpG sites, was applied to assay for DNA methylation[27]. When evaluating the correlation between smoking behaviors and DNA methylation, covariates such as gender and age were adjusted for, along with any relevant technical covariates and blood count measurements[27]. The CpG sites which did not pass the quality control and were available in less than three cohorts were removed, resulting in a total of 485,381 CpG sites were survived in the subsequent meta-analysis[27]. After Bonferroni correction ($P < 1 \times 10^{-7}$), 2623 CpG sites annotated to 1405 genes survived (Supplementary Data 9)[27]. We next extracted mQTLs for these 2,623 CpG sites from the Genetics of DNA Methylation Consortium (GoDMC)[14]. The GoDMC recruited 32,851 European individuals from 36 cohorts to map genetic variants on DNA methylation. To minimize potential confounding and residual variability, the model was adjusted for covariates including sex, age at measurement, batch variables, smoking status, and recorded cell counts[14]. Furthermore, genetic principal components, non-genetic DNA methylation principal components, and predicted

smoking and cell counts were computed as additional confounders according to the GoDMC pipeline[14]. Finally, we identified significant *cis*-mQTL ($P < 1 \times 10^{-8}$, with a distance between of <1MB between mQTL and CpG site) to proxy the level of methylation at each CpG site and additionally performed LD pruning ($r^2 > 0.01$) to select independent IVs for smoking-related DNA methylation. The association estimates between *cis*-mQTL and the risk of CD and UC were derived from the GWAS summary statistics of IBD which has been described above[15]. When investigating the causal impact of genetically determined smoking-related DNA methylation on the risks of CD or UC, each CpG site was considered as exposure variable, and its corresponding mQTLs were employed as instrumental variables. Subsequently, the effect estimates of DNA methylation in each CpG site on per SD increase of CD or UC risk were measured through the Wald ratio approach and combined using the IVW method if there were more than one independent mQTL. The MR-Egger approach was still adopted as a sensitivity analysis to assess potential horizontal pleiotropy[67]. The false discovery rate (FDR) was adopted for multiple testing correction. All MR analyzes were conducted using the "TwoSampleMR" (version 0.5.6) and "MRPRESSO" (version 1.0) R packages within the R software environment (version 4.1.3).

### Genome-wide methylation analysis on CD and UC

Next, we conducted a genome-wide analysis of DNA methylation using a separate cohort to validate the positive findings from the previous step. IBD patients were recruited prospectively from the gastroenterology outpatient clinic across seven centers in Europe, that were part of the IBD-CHARACTER inception cohort (EU reference no 305676)[11]. All cases of IBD met the established diagnostic criteria after a thorough evaluation. Disease clinical phenotypes (i.e., CD and UC) were diagnosed and classified according to the Lennard-Jones, Montreal, and Paris criteria[11]. People who had no symptoms and diagnosis of IBD and healthy participants were included in the control group. A total of 295 controls, 154 CD, and 161 UC patients were included in this analysis[11]. The procedure of genome-wide methylation profiling has been described elsewhere[11]. Simply, the Illumina HumanMethylation450 platform (Illumina, San Diego, CA, USA)[68] was utilized to bisulphite-convert and analyze the peripheral blood leukocyte DNA with a random distribution of cases and controls across chips. We used the R package 'meffil' (R Foundation for Statistical Computing, Vienna)[69] to process data. Samples and probes which did not meet the criteria for analysis were removed. Those samples, which contain more than 1% probes with detection $p$ values less than 0.01 and sex mismatches, were excluded[11]. Probes which meet the following criteria were also removed: bead counts less than 3 or detection $p$ values exceed 0.01 in 10% of samples, containing SNPs with a minor allele frequency ≥0.01 in the European population based on the 1000 Genomes Project[11]. The ComBat function[70] was leveraged to perform batch correction for slide, array, and center, and cell proportions from methylation data were estimated based on the Houseman algorithm[71]. Then, we conducted differentially methylated position analysis and sensitivity analysis with adjustment for age, sex, and cell proportions[72]. To take into account the potential confounding influence associated with the inclusion of patients with unknown smoking status, we also conducted a sensitivity analysis in those patients in whom smoking history was available, and excluding those for whom a smoking history was not taken and performed validation based on the findings of this step.

Since we adopted a stringent $p$-value threshold with FDR correction in the discovery stage and constructed a prior hypothesis for further testing, we set $P < 0.05$ as the statistical significance level in the validation stage. The regulatory effect of DNA methylation on gene expression depends on the location of the CpG probe within a gene. When DNA methylation occurs in promoter regions, it usually inhibits gene expression, whereas when it occurs in gene bodies it promotes gene expression[73].

## Colocalization analysis and the prospective mQTL-smoking interaction analysis

To explore whether the observed associations between CpG site methylation levels and the risk of IBD were influenced by a common causal variant, we conducted a colocalization analysis. We did this for all smoking-related CpG sites meeting one of the two following criteria: 1) regulating the same gene in smoking and IBD-related DNA methylation and acting in the same direction on gene expression, or 2) regulating gene expression in the same direction whilst located at different regions of the gene. For this second criterion, a biological explanation was ensured. To accomplish this, we extracted all accessible mQTLs associated with these CpG sites from GoDMC individually and merged them with GWAS summary-level data for CD or UC. For any CpG site with less than 10 available mQTLs, we excluded it from the colocalization analysis. In colocalization analysis, the posterior probability of five hypotheses was tested: i) H0, indicating no genetic relationship between either trait in this region (PP.H0); ii) H1, suggesting a genetic association only with trait 1 in this region (PP.H1); iii) H2, indicating a genetic association solely with trait 2 in this region (PP.H2); iv) H3, implying correlation between both traits but with different causal variables (PP.H3); and v) H4, suggesting correlation between both traits and sharing a common causal variable (PP.H4)[74]. We considered a summary posterior probability for the CpG site and a posterior probability for the single mQTL ≥80% as convincible colocalization evidence. This analysis was achieved through the "coloc" R package in R software (version 4.1.3)[74].

We additionally performed a prospective gene-environment (G-E) interaction analysis for mQTL-smoking in the UK Biobank through the genotypes of these mQTLs as well as the information for smoking traits (as described above)[64]. Participants with incomplete data were excluded, and 1767 CD cases, 2886 UC cases, and 479,016 controls were finally included for G-E interaction analyzes. To minimize potential confounding factors, the model was adjusted for age at recruitment, sex, level of physical activity, consumption of processed meat, BMI, waist circumference, height, and the first 10 genetic principal components. This analysis was conducted by leveraging the "CGEN" R package (3.34.3)[75]. FDR was employed for multiple testing correction.

### Reporting summary

Further information on research design is available in the Nature Portfolio Reporting Summary linked to this article.

## Data availability

The UK Biobank dataset can be accessed by researchers through a registration and application process at http://ukbiobank.ac.uk/registerapply/, as it serves as an open-access resource for scientific investigations. The UK Biobank application number of this study is 73595. The dataset generated from the UK Biobank to perform Cox regression and G-E interaction analysis are deposited as Source Data for Tables 1, 2, and Supplementary Data 7 in the Figshare Repository (https://doi.org/10.6084/m9.figshare.24791634.v2). The GWAS data for tobacco smoking utilized in this study can be accessed via the GSCAN data portal (https://conservancy.umn.edu/handle/11299/201564). When using the dataset, authors should cite the original publication. The IBD summary statistics can be obtained from the GWAS Catalog under accession codes GCST004132 and GCST004133, respectively. The smoking-SNPs, and smoking-related CpGs used in this study can be found in Supplementary Data 8 and 9, respectively. The mQTLS used in epigenetic MR analysis and colocalization analysis were derived from the GoDMC consortium (http://mqtldb.godmc.org.uk/) by using the code which we have provided. The Source data for Figs. 1, 2, and 3 in this study have also been deposited in the Figshare repository (https://doi.org/10.6084/m9.figshare.24791634.v2). The data used to perform genome-wide DNA methylation analysis cannot be made publicly available on account of data privacy laws and the

national GDPR restrictions affecting some of the collaborating institutions within the IBD-Character Consortium. However, the data can be obtained from the corresponding author upon reasonable request (Jack Satsangi, E-mail: jack.satsangi@ndm.ox.ac.uk.) The request would be responded within one week. When using the data, authors should acknowledge the IBD-Character Consortium and its staff. Source data are provided with this paper.

## Code availability

All software used is publicly available and described in the Methods section of this study. The codes for Cox regression, Mendelian randomization study, genome-wide DNA methylation analysis, and colocalization analysis are deposited in the Github (https://github.com/XueLab157/Smoking-related-DNA-methylation-and-IBD#smoking-related-dna-methylation-and-ibd)[76]. Mendelian randomization analyzes were performed in R based on the 'TwoSampleMR' (version 0.5.6) package (https://mrcieu.github.io/TwoSampleMR/), genetic colocalization analyzes were conducted through the 'coloc' (5.1.0.1) package (https://github.com/chr1swallace/coloc) and visualized using the 'locuscomparer' (1.0.0) package in R (https://github.com/boxiangliu/locuscompare, using default priors).

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

## Acknowledgements

We express our gratitude to all participants and investigators who have contributed to the UK Biobank and published genome-wide association studies for their invaluable data sharing. We also thank the IBD-Character Consortium and its staff for their contribution to this work. The staff list of IBD-Character Consortium is as follow: Adam Carstens, MD, Alan G. Shand, MD, Anette Ocklind, PhD, Anne-Clémence Veillard, PhD, Angelika Merkel, PhD, Anna B. Frengen, PhD, Aina E. F. Moen, PhD, Céline Sabatel, PhD, Charles W. Lees, PhD, Christina Casén, MSc, Christine Olbjørn, MD, Colin L. Noble, MD, Daniel Dirk Repsilber, PhD, Dominique Poncelet, PhD, Ekman, MSc, Daisy Jonkers, PhD, David C. Wilson, MD, MRCPCH, Eddie Modig, BSc, Elaine R. Nimmo, PhD, Erik Andersson, Ian D. Arnott, MD, Erik Pettersson, PhD, Ewa Ciemniejewska, MSc, Ferdinando Bonfiglio, PhD, Fredrik A. Dahl, PhD, Fredrik Hjelm, PhD, Gunn S. Ekeland, Gwo-Tzer Ho, PhD, MSc, Hazel E. Drummond, BSc, Henrik Hjortswang, MD, PhD, Hilde Nilsen, PhD, Ivo G. Gut, PhD, Johan D. Söderholm, MD, PhD, Janne Sølvernes, MS, Kate R. O'Leary, MSc, Leif Törkvist, MD, PhD, Marieke Pierik, MD, PhD, Mats Gullberg, PhD, Marta Gut, PhD, Mauro D'Amato, PhD, Mårten Lindqvist, PhD, Monica Bayes, PhD, Mikael Sundell, BSc, Niklas Nordberg, PhD, Nicholas A. Kennedy, MBBS, PhD, FRACP, Nicholas T Ventham, MRCS[Eng] PhD, MBBS, Panpan You, MS, Ray K. Boyapati, MD, Renaud Schoemans, PhD, Simon C. Heath, PhD, Torbjørn Lindahl, MSc, Tone M. Tannæs, PhD, Trond Espen Detlie, MD. This study is supported by funding that are displayed as follow: X.L.: The Natural Science Fund for Distinguished Young Scholars of Zhejiang Province (LR22H260001), the National Nature Science Foundation of China (82204019), Healthy Zhejiang One Million People Cohort (K-20230085). E.T.: CRUK Career Development Fellowship (C31250/A22804). Z.L.: The National Nature Science Foundation of China (82370532). J.W.: The Research Foundation Flanders (FWO), Belgium by a PhD Fellowship strategic basic research (SB) grant (1SO6023N). J.K.N.: The National Science Centre (Poland), 2020/39/D/NZ5/02720.

## Author contributions

X.L., J.S., E.T., and Z.L. designed and supervised the study. H.Z., R.K., and J.C. participated in the data curation. H.Z., R.K., and J.C. performed the data analyzes and prepared the tables and figures. J.S, E.T, Z.L, J.Z, X.Z, A.A, A.N, N.V, J.W, G.T.H, M.D, and J.K.N contributed to analysis strategy and data. H.Z. and R.K. wrote the original draft. Y.D. and X.L. contributed to project administration. All authors critically revised the content and contributed to editing the paper. X.L. is the study guarantor. The corresponding author (X.L.) attests that all listed authors meet authorship criteria and that no others meeting the criteria have been omitted.

## Competing interests

J.K.N. declares to be funded by a grant from the Biocodex Microbiota Foundation. The remaining authors declare no competing interests.
