## [Peer Review File · Nature Communications]

Altered DNA methylation within DNMT3A, AHRR, LTA/TNF loci mediates the effect of smoking on inflammatory bowel diseaseREVIEWER COMMENTS

Reviewer #1 (Remarks to the Author):

This is an interesting manuscript that attempts to tackle the difficult task of understanding the dynamic interplay of smoking and the occurrence of inflammatory bowel disease. Since the manuscript is almost completely focused upon genetic and DNA methylation related data, I suggest a change the title to reflect this (i.e. DNA methylation rather than 'epigenetic') – 'epigenetic' can refer to many other types of data and different biology.

INTRODUCTION AND RESULTS

The authors use a genetic epidemiologic approach to assess the effect of smoking on Crohn's disease (CD) and Ulcerative Colitis (UC). The initial analysis of the UK Biobank suggests that smoking confers risk in both CD and UC, albeit with a somewhat attenuated risk for UC in current smokers. A curious finding that is not commented upon by the authors is found in Suppl Table 6 where the data are stratified by age of onset. There was a statistically significant inverse association for BOTH UC and CD afforded by smoking cessation after age 50 in the young (<60) onset disease. The magnitude of the protection was quite large (larger than the risk estimates in the overall analysis) and would merit mention and discussion (and perhaps inclusion of the Table in the body of the paper). There is debate about the differences between early and elderly onset of these conditions and this finding may offer some insight into this issue (but one speculative idea is below – others may also deserve mention).

The analyses of the cis-mQTLs are conducted using yet another data set – that of de Lange et al (Nat Genet 49, 2017). In an isolated reading of the results this fact would elude the reader. The reference should be in the Results as well as Methods and source attributed (e.g. European Genome-phenome Archive). This will make understanding the flow of the paper simpler.

The final replication analysis is inadequately described; nowhere is the demographic character of the IBD-CHARACTER cohort described and this is important for the interpretation of the data. For example, this is not a large cohort and it is unclear how many smokers were included in it. Table 2 is quite poorly described; it is not clear even what comparison is being made (and hence where the p-value comes from). The beta value is presumably the difference in methylation from the 450K array (this is nowhere stated) but what is the comparison? The reference to the original study in the Table might profitably include the first authors last name (they use only the first name) and the PMID. The text (p 12 lines 247-251) states that two CpGs had a "consistent effect on the regulation of gene expression (Table 2)." I find no data referent to gene expression in Table 2 (or anywhere else). What data is this statement referring to?

DISCUSSION

It is known that smokers also have distinct immune profiles (see Wang et al, Clin Epigenetics, 2023) and that different leukocyte subtypes are differentially subject to methylation alteration by smoking. Development of the mQTLs and the SNP approach used could be sensitive to differences in immune cell profile. Further, since there are some CpGs where the methylation associated with smoking is 'reversible' and some CpGs where the methylation is refractory to quitting smoking (e.g., 'persistent') (also see Guida et al, Hum Mol Genet 24, 2015), the initial discovery of mQTLs may be biased by the characteristics of the source population. Indeed, the 'persistent' changes may represent stem cell effects which could have implications for this study (note that cg23916896, associated with AHRR, is such a 'persistent' loci). Certainly, the estimates for effect size could differ for each type of CpG, dependent on smoking behavior. It may be that the kinetics (and dynamics) of DNA methylation associated with smoking make the detection of mQTLs more difficult; this may mean that the current data underestimate the importance of smoking in IBD.

The Discussion also states that Piovani et al (2019) reported "an inverse association between current smoking and UC risk...however, our study found no supportive evidence on this". I believe that this is

an overstatement. Table S6 shows an inverse point estimate for current smokers in the younger onset group (NS) and, as noted above, the younger recent ex-smokers have a significant inverse association. Is it not possible that smoking-related shifts in B cell populations that revert in recent quitters (something that may not happen in the elderly due to their paucity of naïve cells that could make reversion not possible) provide a window of protection for onset of this disease? This is mere speculation, of course, but it is added here as a caution against use of the current language which is perhaps too strong. I suggest revision and inclusion of the fact that cell subtype shifts are known to be associated with smoking and subsequent immune perturbations in the Discussion.

OTHER

I suggest that the authors revise some of the text with attention to grammar. The first sentence in the Abstract, for example, is incomplete. The second sentence is missing the article 'a'. Additional examples of poor language usage exist.

Reviewer #2 (Remarks to the Author):

In the study, authors assessed the association between cigarette smoking and IBD risk through a prospective cohort study in the UK Biobank. Then, authors evaluated the causal association of smoking behaviors and smoking-related DNA methylation with IBD by performing MR analysis, and finally validated the associations via genome-wide methylation and colocalization analyses. This is a comprehensive study, and I have two main questions:

1. For the integration analysis of multiple omics data, has the author considered batch effects for each type of omics data internally and between different types of omics data? How does the author perform batch effect correction between the different data sets?
2. Was the author's research targeted towards a specific ethnic group? How can the research results be generalized to multiple ethnic groups or populations?

If satisfactory explanations can be provided for the above two questions, I suggest that this article can be published after revision.

Reviewer #3 (Remarks to the Author):

This manuscript by Zhang et al. investigated the potential joint associations between smoking, DNA methylation and inflammatory bowel disease (IBD) by leveraging the large cohort from UK biobank and other public available data sources. The authors conducted cox regression to infer the risk of smoking on Crohn's disease (CD) and ulcerative colitis (UC), and then performed two mendelian randomization analyses to infer the causality between smoking and CD&UC as well as between DNA methylation and CD&UC. The authors further put all different pieces together and applied a colocalization analysis to validate the links among the genetic risk, smoking, DNA methylation and IBD disease. Overall, this study demonstrates a well-designed research approach and the manuscript is adequately written. However, I have noticed a dearth of specific information in certain sections, particularly the methodology section. Specific comments are as follows:

1. The detailed screening process and criteria of included participant should be provided along with the data sources in Figure 1 A. In addition, I feel like the labels (A-E) are not necessary as they were not referred in the main-text or even in the figure legend after all.
2. It is interesting the authors performed stratification analysis in two age groups in the prospective cohort study. Would it be possible to repeat the stratification analyses in MR analyses? It would be also interesting to see the results after sex/gender stratifications?
3. It is certainly interesting to apply the MR model to infer the associations between smoking and the

risk of IBD to explore smoking-related biological pathways in the pathogenesis of IBD. Meanwhile, given the gene-environmental interactions were reported to be involved in the IBD disease development, it would be interesting to check whether most of mQTLs were related to SNP-smoking interactions.

4. Line 181, the authors argued to use nominal P-value <0.05 as the significant threshold in a replication study. It would be beneficial if the authors acknowledge in the discussion that their chosen nominal p-value threshold of 0.05 is more liberal than usual. Furthermore, providing a detailed explanation for why they believe this is warranted in their study. The authors should discuss the potential limitation when using more stringent thresholds such as FDR corrected thresholds.

5. Line 175-178 & Line 241-244, I understand the DNA methylation data of IBD EWAS might be prepared in another study as indicated in Table 2 (Rahul et al or actually should be Kalla et al). But the details of the dataset, the detailed process etc. should be included here if the study was not published or should be cited properly if already published. Particularly, it is not clear how many samples/probes were filtered at each step due to p-value filtering, SNP probe removal and cross-hybridising etc. Moreover, how the authors reached the final sample size $n=401$ and $n=428$ in Table S11 and Table S12?

6. If I understood correctly, Table S11 and Table S12 are the DMP results for CD vs control and UC vs control, respectively; and Table 2 are the smoking-associated CpGs which showing significant associations with risk of CD or UC and were further replicated in the IBD-CHARACTER cohort (Table S11 or Table S12). However, several Table 2 CpGs were not replicated/reported in Table S11 or Table S12. Please add more details to clarify the Table 2 and its relationship to Table S11 and Table S12.

7. Line 250-251, how the authors come to the statement that cg03599224 and cg16280132 had a consistent effect on the regulation of gene expression from Table 2? Any other data analysis was missing here to support this result?

8. It is highly recommended to add the methylation difference between IBD and control in Table 2 or Table S11 and S12.

9. Before narrowing down the smoking-related CpG sites by the replication study, did the author explore any biological implications of the genes associated with the 46 and 69 smoking-CpGs which also show associations with the risk of CD and UC?

10. It is suggested that the authors add detailed legends to Figures or Tables. For instance, Figure 4 has three panels but none of them were explained in the legend. What are the headers (e.g. PP.H*) of Table S13 etc.? I would also suggest the authors to add the detailed number of mQTLs for each CpG in Table S13. Moreover, we can guess LCI and UCI in Table 1 but actually these were not indicated/used in the table, so they could be deleted in Table 1. There seems a typo for the p-value of the Crude model for CD in the last row of the Table 1.

11. It is very hard to read the Table S9 and Table S10, I would suggest to move them to excel tables like Table S3, S4 etc.

12. Line 213, what is the "genetically predicted smoking behaviors"? why the "predicted" behaviors were introduced here?

REVIEWER COMMENTS

Reviewer #1 (Remarks to the Author):

This is an interesting manuscript that attempts to tackle the difficult task of understanding the dynamic interplay of smoking and the occurrence of inflammatory bowel disease. Since the manuscript is almost completely focused upon genetic and DNA methylation related data, I suggest a change the title to reflect this (i.e. DNA methylation rather than 'epigenetic') – 'epigenetic' can refer to many other types of data and different biology.

Response: We appreciate the reviewer's objective evaluation of our work and the helpful comment to improve this study. We agree that the title should reflect the contents as specifically as possible. Therefore, the word "epigenetic" has been replaced by "DNA methylation" in the title.

Title, Line 1-2: *"Altered DNA methylation within DNMT3A, AHRR, LTA/TNF loci mediates the effect of smoking on inflammatory bowel diseases"*.

INTRODUCTION AND RESULTS

The authors use a genetic epidemiologic approach to assess the effect of smoking on Crohn's disease (CD) and Ulcerative Colitis (UC). The initial analysis of the UK Biobank suggests that smoking confers risk in both CD and UC, albeit with a somewhat attenuated risk for UC in current smokers. A curious finding that is not commented upon by the authors is found in Suppl Table 6 where the data are stratified by age of onset. There was a statistically significant inverse association for BOTH UC and CD afforded by smoking cessation after age 50 in the young (<60) onset disease. The magnitude of the protection was quite large (larger than the risk estimates in the overall analysis) and would merit mention and discussion (and perhaps inclusion of the Table in the body of the paper). There is debate about the differences between early and elderly onset of these conditions and this finding may offer some insight into this issue (but one speculative idea is below – others may also deserve mention).

Response: Thanks for this comment. Based on this and other comments, we have moved Supplementary Table 6 into the main body as Table 2. In addition, we discussed these findings in the Results and Discussion sections of the paper. Please find the details in Table 2 and the Results and Discussion sections

highlighted in yellow.

Results, Line 109-112: *“In addition, smoking cessation after 50 years old was inversely associated with the risk of CD (HR: 0.12, 95%CI: 0.02-0.96, P=0.045) and UC (HR: 0.25, 95%CI: 0.09-0.74, P=0.012) in those with disease onset less than 60 years of age (Table 2).”*

Discussion, Line 179-181: *“Interestingly, smoking cessation after the age of 50 was associated with decreased risks of both CD and UC in those presenting within 10 years of cessation.”*

Discussion, Line 212-228: *“Of interest, we observed a benefit of smoking cessation after 50 years of age on both CD and UC risk in those under 60 years but not in those older than 60. This protective effect may be due to the fact that smoking cessation might allow for recovery of the DNA methylation levels of smokers.^{26,27} Other immunological mechanisms associated with ageing may also be relevant in the observed age-related changes, including differential effects on memory B cells.²⁸ Of particular note is the fact that the effect of smoking cessation was seen both in CD as well as UC, and this is apparently at odds with the well-evidenced and accepted association of smoking with susceptibility to CD in young adults.^{29,30} The latter is of interest in the light of emerging data that the phenotypic presentation, as well as complication profile and response to therapy seems to differ between early-onset and late-onset IBD.³¹⁻³³ The current observations provide further evidence that there may be age-related heterogeneity in disease pathogenesis. Thus, aetiological factors involved in late-onset disease may differ from those in disease with onset in childhood or early-adulthood, a fact which may be important in designing prevention strategies.”*

The analyses of the cis-mQTLs are conducted using yet another data set – that of de Lange et al (Nat Genet 49, 2017). In an isolated reading of the results this fact would elude the reader. The reference should be in the Results as well as Methods and source attributed (e.g. European Genome-phenome Archive). This will make understanding the flow of the paper simpler.

Response: Many thanks for this helpful suggestion. We have added the reference to the Results and Methods. The details are highlighted in yellow in

the article.

Methods, Line 457-460: *“The association estimates between cis-mQTL and the risk of CD and UC were derived from the GWAS summary-level data of IBD which has been described above.¹⁵”*

Results, Line 121-123: *“Among the 2,623 CpG sites that associate with smoking behaviors,¹⁴ 1,938 for CD and 1,939 for UC had available cis-mQTLs that could be used as proxies of smoking-related DNA methylation in the two-sample MR analyses.¹⁵”*

The final replication analysis is inadequately described; nowhere is the demographic character of the IBD-CHARACTER cohort described and this is important for the interpretation of the data. For example, this is not a large cohort and it is unclear how many smokers were included in it. Table 2 is quite poorly described; it is not clear even what comparison is being made (and hence where the p-value comes from). The beta value is presumably the difference in methylation from the 450K array (this is nowhere stated) but what is the comparison? The reference to the original study in the Table might profitably include the first authors last name (they use only the first name) and the PMID. The text (p 12 lines 247-251) states that two CpGs had a “consistent effect on the regulation of gene expression (Table 2).” I find no data referent to gene expression in Table 2 (or anywhere else). What data is this statement referring to?

Response: Thank you very much for these valuable comments. The genome-wide DNA methylation analysis procedures as well as the demographic character of the IBD-CHARACTER cohort have been published by Kalla et al.[1] To avoid overlap, we have described the main procedures of the study in the Supplementary Methods and shown the specific demographic characteristics in Supplementary Table 6 and described these demographic characteristics in the Results section. In addition, it is worth noting that these genetic loci remain differentially methylated after sensitivity analysis which excluded those patients for whom data regarding smoking status was not available. Please find the details and changes in the later section of the response to this comment.

In the original Table 2, we compared the findings of epigenetic MR with genome-

wide DNA methylation analysis and identified the common gene loci which are associated with both smoking- and IBD-related DNA methylation. We only displayed the common genetic loci, and provided its causal effect on IBD which was found in the epigenetic MR analysis. We have now displayed the findings from both the epigenetic MR and genome-wide DNA methylation profiling, and re-ordered them as Table 3 and Table 4. The beta value of epigenetic MR represents the magnitude of causal effect per unit increase in genetically proxied DNA methylation at corresponding CpG site on the risk of IBD, whereas the beta value of genome-wide DNA methylation analysis represents the difference in beta values at each CpG site between disease and controls. As the beta values represent different effect estimates, they are not comparable, whilst the statistical significance of their p-values for these differentially methylated positions (DMPs) was referred for validation. To avoid misunderstanding, we have now added Table 4 to display the validated genetic loci in IBD EWAS. As for the regulation of DNA methylation on gene expression, we discuss this in the context of the potential transcriptional effect of DNA methylation on the corresponding gene based on the position of the corresponding CpG site. Generally, different locations (i.e., gene body vs the promoter) of DNA methylation in genes have opposite effects on gene expression. It usually inhibits the gene expression when DNA methylation occurs in the promoter region, while it promotes gene expression when occurring in the gene body.[2] For clarification, we have added this to the Methods.

According to the comments, we also have supplemented the first author's last name and the PMID of the original study in Table 4. Details of the changes can be found below and the section highlighted in yellow in the paper.

Methods, Line 486-490: *“The regulatory effect of DNA methylation on gene expression depends on the location of the CpG probe within a gene. When DNA methylation occurs in promoter regions, it usually inhibits gene expression, whereas when it occurs in gene bodies it promotes gene expression.”⁷¹*

Results, Line 140-150: *“We additionally performed a genome-wide methylation analysis on CD and UC in the IBD-CHARACTER inception cohort to validate the association of altered DNA methylation at genetic loci related to*

smoking exposure. There were 343 IBD patients and 295 controls in the cohort, and 154 patients were classified as CD, 161 as UC, and 28 as unspecified IBD. There were 54 current and 103 former smokers in the IBD group. Smoking exposure details are provided in Supplementary Table 6. The mean age of IBD patients was 34 years [range 7-79], and 33 years [range 3-79] in controls,¹¹ respectively (Supplementary Table 6). The differentially methylated CpG sites and their related loci are displayed in Supplementary Tables 7 and 8; and the results of sensitivity analysis are presented in Supplementary Tables 9 and 10

Supplementary Table, Table S6: “*The demographic characteristics of the IBD-CHARACTER cohort.*”

DISCUSSION

It is known that smokers also have distinct immune profiles (see Wang et al, Clin Epigenetics, 2023) and that different leukocyte subtypes are differentially subject to methylation alteration by smoking. Development of the mQTLs and the SNP approach used could be sensitive to differences in immune cell profile. Further, since there are some CpGs where the methylation associated with smoking is ‘reversible’ and some CpGs where the methylation is refractory to quitting smoking (e.g., ‘persistent’) (also see Guida et al, Hum Mol Genet 24, 2015), the initial discovery of mQTLs may be biased by the characteristics of the source population. Indeed, the ‘persistent’ changes may represent stem cell effects which could have implications for this study (note that cg23916896, associated with AHRR, is such a ‘persistent’ loci). Certainly, the estimates for effect size could differ for each type of CpG, dependent on smoking behavior. It may be that the kinetics (and dynamics) of DNA methylation associated with smoking make the detection of mQTLs more difficult; this may mean that the current data underestimate the importance of smoking in IBD.

Response: Many thanks for this insightful comment and the helpful references you provided. We have considered the cited references suggested. We agree with the points made by the referee with respect to the need to discuss the potential confounding effects of alterations in cell type proportions over time, and this is a consideration of a potential confounder in a cross-sectional study that we acknowledge. Similarly, the second point relating to kinetics and dynamics of DNA methylation which may lead to an under estimation of the

effect of smoking is pertinent. We have altered the manuscript to take into account these helpful observations and have included these references. Details are as follows.

Discussion, Line 351-365: *“In addition to statistical and analytical considerations, there are also biological issues to consider when interpreting these data. Changes in methylation are cell type specific, and immune cell subtypes may potentially alter over time due to smoking, or disease activity, or drug therapy; and this may affect the observed smoking-related changes in DNA methylation and transcription.⁶⁵ Since the smoking-related DNA methylation in CpG sites was obtained from a cross-sectional EWAS, we were unable to evaluate the time course, kinetics and dynamics of cigarette smoking on DNA methylation. Further longitudinal studies to explore the causal effect of smoking on DNA methylation over time are warranted. It is reported that methylation at some CpG sites reverts to some extent after smoking cessation, whereas some CpG sites remain differentially methylated even after smoking cessation >35 years later.²⁶ In this context, it is possible that the initial identification of mQTLs may have been affected by the characteristics of the study population, and we may underestimate the importance of smoking.”*

Discussion, Line 375-380: *“Finally, whilst we demonstrated robust and validated data, we have concentrated on epigenetic signals in the blood and the EWAS of smoking-related DNA methylation was conducted by using blood samples. It should be considered that the DNA methylation signature is diverse in different tissues and immune cell subtypes,²⁸ further research using data derived from intestinal tissues and different immune cell subtypes is warranted.”*

The Discussion also states that Piovani et al (2019) reported “an inverse association between current smoking and UC risk...however, our study found no supportive evidence on this”. I believe that this is an overstatement. Table S6 shows an inverse point estimate for current smokers in the younger onset group (NS) and, as noted above, the younger recent ex-smokers have a significant inverse association. Is it not possible that smoking-related shifts in B cell populations that revert in recent quitters (something that may not happen in the elderly due to their paucity of naïve cells that could make reversion not possible) provide a window of protection for onset of this disease? This is mere

speculation, of course, but it is added here as a caution against use of the current language which is perhaps too strong. I suggest revision and inclusion of the fact that cell subtype shifts are known to be associated with smoking and subsequent immune perturbations in the Discussion.

Response: Many thanks for this comment. Indeed, an inverse association (not significant) between current smoking and UC risk has been observed in those under 60 years, and smoking cessation was associated with a decreased risk of CD and UC in the younger onset population. We agree with the need to discuss further and consider that smoking cessation may affect the DNA methylation levels of smokers[3, 4] and that differential smoking-related alterations in B cell populations may be relevant. We have revised the Results and Discussion according to the comments.

Results, Line 106-112: *“Albeit not significant, a trend towards a reverse association pattern between current smoking and UC risk was observed in those under 60 years (HR: 0.82, 95%CI: 0.61-1.10, P=0.179). In addition, smoking cessation after 50 years old was inversely associated with the risk of CD (HR: 0.12, 95%CI: 0.02-0.96, P=0.045) and UC (HR: 0.25, 95%CI: 0.09-0.74, P=0.012) in those with disease onset less than 60 years of age (Table 2).”*

Discussion, Line 207-209: *“The authors reported an inverse association between current smoking and UC risk (OR:0.58, 95%CI: 0.45-0.75); our study also found a similar trend in the population under 60, albeit not statistically significant.”*

Discussion. Line 212-228: *“Of interest, we observed a benefit of smoking cessation after 50 years of age on both CD and UC risk in those under 60 years but not in those older than 60. This protective effect may be due to the fact that smoking cessation might allow for recovery of the DNA methylation levels of smokers.^{26,27} Other immunological mechanisms associated with ageing may also be relevant in the observed age-related changes, including differential effects on memory B cells.²⁸ Of particular note is the fact that the effect of smoking cessation was seen both in CD as well as UC, and this is apparently at odds with the well-evidenced and accepted association of smoking with susceptibility to CD in young adults.^{29,30} The latter is of interest in the light of emerging data that the phenotypic presentation, as well as complication profile*

and response to therapy seems to differ between early-onset and late-onset IBD.³¹⁻³³ The current observations provide further evidence that there may be age-related heterogeneity in disease pathogenesis. Thus, aetiological factors involved in late-onset disease may differ from those in disease with onset in childhood or early-adulthood, a fact which may be important in designing prevention strategies.”

OTHER

I suggest that the authors revise some of the text with attention to grammar. The first sentence in the Abstract, for example, is incomplete. The second sentence is missing the article ‘a’. Additional examples of poor language usage exist.

Response: Thanks for this helpful comment to improve our article. We have revised the grammar carefully in the paper. Detailed changes can be found in the highlighted section of the manuscript. Here are some examples.

Abstract, Line 37-38: *“This study aimed to explore biological pathways by which smoking may be involved in the pathogenesis of inflammatory bowel disease (IBD).”*

Abstract, Line 38-41: *“We first conducted a prospective cohort study and a Mendelian randomization study to examine the association of smoking behaviors and related DNA methylation with Crohn’s disease (CD) and ulcerative colitis (UC) risks.”*

Abstract, Line 49-51: *“Our study identifies biological mechanisms and pathways involved in the effects of smoking in the development of IBD.”*

Introduction, Line 66-68: *“Despite decades of research using different methodologies, our current understanding of the mechanisms underlying these effects remain limited.”*

Discussion, Line 182-185: *“while cg03599224 (located near LTA/TNF within the HLA class III locus) was associated with CD risk, and cg14647125 and cg23916896 [all located in AHRR] were linked to UC risk.”*

Reviewer #2 (Remarks to the Author):

In the study, authors assessed the association between cigarette smoking and IBD risk through a prospective cohort study in the UK Biobank. Then, authors evaluated the causal association of smoking behaviors and smoking-related DNA methylation with IBD by performing MR analysis, and finally validated the associations via genome-wide methylation and colocalization analyses. This is a comprehensive study, and I have two main questions:

Response: Many thanks for your positive evaluation of this study. All authors are grateful for the reviewer's time and efforts that have been put into helping to improve this paper. We address the received comments very carefully and append a point-by-point response to each of these comments in the remainder of this letter.

1. For the integration analysis of multiple omics data, has the author considered batch effects for each type of omics data internally and between different types of omics data? How does the author perform batch effect correction between the different data sets?

Response: Thank you for this important comment. In the prospective cohort study in the UK Biobank, the exposure and outcome were ascertained according to the questionnaires and ICD codes, respectively. Therefore, there are no batch effects that needed to be considered. In the original GWAS of smoking and IBD and the EWAS of smoking, researchers have corrected for any batch effects.[4-6] The five sections of this study were conducted independently to validate or explain the results of the previous analysis stage by stage, there was no intention to pool these data together for analysis. Therefore, there is no expectation for batch effect that could bias the results.

2. Was the author's research targeted towards a specific ethnic group? How can the research results be generalized to multiple ethnic groups or populations?

Response: Thank you very much for this comment. As mentioned in the Strengths and Limitations section, our study mainly focused on European ancestry, which is both a potential strength as well as a potential limitation. On one hand, focusing on a specific ethnic group (i.e., European) may have reduced potential confounding caused by population structure. On the other

hand, it limits the generalizability of our findings to other ethnic populations. Nevertheless, our study can provide a new perspective and paradigm for studying the mechanisms of smoking causing IBD. If and when, GWAS and EWAS data on other ethnicities become available in the future, these findings may be validated in other ancestries. We have added this comment to the limitations section. Details are as follows.

Discussion, Line 334-339: *“Our study mainly focused on the European population, which has allowed us to minimize any effect of confounding caused by population structure. However, in turn, this aspect also limits the generalizability of our findings to other ethnic populations. Once GWAS and EWAS data on other ethnicities become available, these findings will be explored.”*

Reviewer #3 (Remarks to the Author):

This manuscript by Zhang et al. investigated the potential joint associations between smoking, DNA methylation and inflammatory bowel disease (IBD) by leveraging the large cohort from UK biobank and other public available data sources. The authors conducted cox regression to infer the risk of smoking on Crohn's disease (CD) and ulcerative colitis (UC), and then performed two mendelian randomization analyses to infer the causality between smoking and CD&UC as well as between DNA methylation and CD&UC. The authors further put all different pieces together and applied a colocalization analysis to validate the links among the genetic risk, smoking, DNA methylation and IBD disease. Overall, this study demonstrates a well-designed research approach and the manuscript is adequately written. However, I have noticed a dearth of specific information in certain sections, particularly the methodology section.

Response: We are very grateful for your positive evaluation of this study. All authors appreciate the reviewer's time and efforts that have been put into helping to improve this paper. We addressed the received comments very carefully and append a point-by-point response to each of these comments in the remainder of this letter.

Specific comments are as follows:

1. The detailed screening process and criteria of included participant should be provided along with the data sources in Figure 1 A. In addition, I feel like the labels (A-E) are not necessary as they were not referred in the main-text or even in the figure legend after all.

Response: Thanks for this helpful suggestion. We have revised Figure 1 according to the comments. Details are as follows.

“Figure 1. The schematic presentation of study design. CD, Crohn’s disease. UC, ulcerative colitis. SNPs, single nucleotide polymorphisms. MR, Mendelian Randomization. GWAS, genome-wide association study. mQTL, methylation quantitative trait loci. IBD, inflammatory bowel disease. DMP, differentially methylated position. EWAS, epigenome-wide association study. FDR, false discovery rate.”

2. It is interesting the authors performed stratification analysis in two age groups in the prospective cohort study. Would it be possible to repeat the stratification analyses in MR analyses? It would be also interesting to see the results after sex/gender stratifications?

Response: Many thanks for proposing this interesting suggestion. The MR analysis is limited by the availability of the summary-level data on CD and UC. Since there were no such sex- and age-stratified summary-level data available for CD and UC, we were unfortunately not able to do these stratification analyses.

Discussion, Line 365-368: “As there were no sex- and age-stratified summary-level data available for CD and UC, we were unable to conduct this stratified MR analysis. These MR analyses could be added when specific summary-level data would become available in the future.”

3. It is certainly interesting to apply the MR model to infer the associations between smoking and the risk of IBD to explore smoking-related biological

pathways in the pathogenesis of IBD. Meanwhile, given the gene-environment interactions were reported to be involved in the IBD disease development, it would be interesting to check whether most of mQTLs were related to SNP-smoking interactions.

Response: We also think that it is of interest to explore whether any gene-environment (G-E) interaction effects exist between the mQTLs and smoking phenotypes (i.e., smoking status, pack years of smoking, and age stopped smoking). Therefore, we have additionally performed a G-E interaction analysis in the UK Biobank for those mQTLs corresponding to the CpG sites, which were validated in the replication stage. However, we did not find any mQTLs that interacted with the smoking phenotypes. Therefore, we did not go on to perform a further stratification analysis. Please find the details in the Table S10 and the highlighted sections in the text.

Method, Line 513-523: *“We additionally performed a prospective mQTL-smoking interaction analysis by obtaining the genotypes of mQTLs of these CpG sites as well as the baseline information of three smoking traits (i.e., smoking status, pack years of smoking, and age stopped smoking) in the UK Biobank.⁶⁷ Participants with incomplete data were excluded, and 1767 CD cases, 2886 UC cases, and 479,016 controls were finally included for gene-environment interaction analyses. Age at recruitment, sex, physical activity, processed meat consumption, BMI, waist circumference, height, and the first ten genetic principal components were adjusted for potential confounding in the model. This analysis was conducted by leveraging the “CGEN” R package (3.34.3).⁷³ FDR was employed for multiple testing correction.”*

Results, Line 172-175: *“We did not identify any mQTL that significantly interacted with all these three smoking phenotypes after FDR correction (Supplementary Table 12). Therefore, we did not conduct any further stratification analysis.”*

4. Line 181, the authors argued to use nominal P-value <0.05 as the significant threshold in a replication study. It would be beneficial if the authors acknowledge in the discussion that their chosen nominal p-value threshold of 0.05 is more liberal than usual. Furthermore, providing a detailed explanation for why they believe this is warranted in their study. The authors should discuss

the potential limitation when using more stringent thresholds such as FDR corrected thresholds.

Response: Thanks for this insightful and important comment. In this study, we aimed to comprehensively evaluate the association of smoking with the risk of CD and UC and explore the potential biological pathway by which smoking may be involved in the pathogenesis of IBD. As we did not propose any prior hypothesis in the discovery stage, we therefore applied a corrected p-value threshold by FDR to account for multiple testing. In the validation stage, we had constructed a specific prior hypothesis based on findings from the discovery stage, for which a norm p-value threshold (<0.05) was deemed acceptable. However, to address this comment adequately, we adopted the FDR correction for the number of methylation signals that were taken forward for the validation in EWAS of CD and UC, respectively. Specifically, 21 (corresponding to 65 CpG sites) and 35 (corresponding to 135 CpG sites) genetic loci were taken forward for replication in the EWAS of CD and UC, respectively. As shown in the attached table, when a more stringent FDR correction was applied in the replication stage, all these genetic loci remain significant (the FDR-corrected p-value <0.05). Therefore, those findings are robust regardless of whether multiple test correction was performed. To address this properly, we have now added the following explanation and discussion in the manuscript.

Methods, Line 484-486: *“Since we adopted a stringent p-value threshold with FDR correction in the discovery stage and constructed a prior hypothesis for further testing, we set $P<0.05$ as the statistical significance level in the validation stage.”*

Discussion, Line 344-350: *“A more liberal significance threshold was used in the genome-wide methylation profiling, which is based on a prior hypothesis regarding the loci involved, rather than a threshold associated with hypothesis-free analysis for those genetic loci identified in the epigenetic MR analysis. Furthermore, it is important to note that all findings remained significant when FDR correction was conducted for those validated signals (65 for CD and 135 for UC).”*

Table 4. Validation of the genetic loci in the genome-wide methylation analysis of Crohn's disease and ulcerative colitis.

Outcome	Gene	Genome-wide methylation analysis						
		CpG site	Region	beta	se	P-value	FDR	
Crohn's disease	LTA	cg16280132	5'UTR	0.009	0.004	3.43×10 ⁻²	4.49×10 ⁻²	
		cg23393100	body	0.002	0.001	1.46×10 ⁻²	3.40×10 ⁻²	
	DNMT3A	cg17207266	body	0.008	0.002	4.36×10 ⁻⁵	1.42×10 ⁻³	
		cg05544807	body	0.012	0.003	2.77×10 ⁻⁴	3.61×10 ⁻³	
		cg20702417	body	0.010	0.003	7.66×10 ⁻⁴	7.11×10 ⁻³	
		cg03314052	body	0.008	0.003	1.69×10 ⁻²	3.43×10 ⁻²	
		cg11779362	body	0.002	0.001	1.03×10 ⁻²	3.04×10 ⁻²	
		cg00970057	body	0.009	0.003	3.59×10 ⁻³	1.56×10 ⁻²	
	HIVEP3	cg09501963	body	0.005	0.002	2.83×10 ⁻²	4.34×10 ⁻²	
		cg24051040	body	0.004	0.002	3.87×10 ⁻²	4.49×10 ⁻²	
		cg00326958	5'UTR	-0.007	0.003	8.06×10 ⁻³	3.00×10 ⁻²	
	MACROD1	cg10612274	body	0.025	0.007	2.06×10 ⁻⁴	3.35×10 ⁻³	
		cg17590162	body	0.013	0.005	1.30×10 ⁻²	3.20×10 ⁻²	
		cg12424921	body	0.010	0.005	2.84×10 ⁻²	4.34×10 ⁻²	
		cg04582871	body	0.011	0.005	3.77×10 ⁻²	4.49×10 ⁻²	
		cg09908110	body	0.001	0.001	4.56×10 ⁻²	4.69×10 ⁻²	
		cg09375205	body	0.006	0.002	1.56×10 ⁻²	3.43×10 ⁻²	
		cg02374207	body	0.002	0.001	1.12×10 ⁻²	3.12×10 ⁻²	
		Ulcerative colitis	AHRR	cg22356527	body	-0.005	0.002	3.04×10 ⁻²
	cg19602050			body	-0.001	0.000	1.72×10 ⁻²	3.59×10 ⁻²
cg12806681	body			0.006	0.002	5.53×10 ⁻³	2.71×10 ⁻²	
cg02088390	body			-0.002	0.001	2.33×10 ⁻²	4.37×10 ⁻²	
cg26703534	body			0.013	0.004	1.22×10 ⁻³	1.25×10 ⁻²	
cg11894422	body			0.005	0.002	3.21×10 ⁻²	4.47×10 ⁻²	
cg21161138	body			0.013	0.004	3.52×10 ⁻³	2.16×10 ⁻²	
cg23576855	body			0.034	0.016	3.71×10 ⁻²	4.47×10 ⁻²	
cg11557553	body			-0.004	0.002	4.66×10 ⁻²	4.87×10 ⁻²	
cg25648203	body			0.010	0.004	1.51×10 ⁻²	3.42×10 ⁻²	
DNMT3A	cg26544247		body	0.006	0.003	3.71×10 ⁻²	4.47×10 ⁻²	
	cg20702417		body	0.006	0.003	4.85×10 ⁻²	4.93×10 ⁻²	
	cg15843262		body	0.005	0.002	5.10×10 ⁻³	2.65×10 ⁻²	
	cg17207266		body	0.006	0.002	4.35×10 ⁻³	2.45×10 ⁻²	
	cg27369452		body	0.001	0.001	8.24×10 ⁻³	3.05×10 ⁻²	
	cg05544807		body	0.009	0.003	2.81×10 ⁻³	2.00×10 ⁻²	
	cg11779362		body	0.002	0.001	4.91×10 ⁻²	4.93×10 ⁻²	

5'UTR, 5' untranslated region. Body, Gene body. These loci were validated by a genome-wide DNA methylation analysis of inflammatory bowel disease (Kalla, et al, *J Crohns Colitis*. PMID: 36029471).

5. Line 175-178 & Line 241-244, I understand the DNA methylation data of IBD EWAS might be prepared in another study as indicated in Table 2 (Rahul et al or actually should be Kalla et al). But the details of the dataset, the detailed process etc. should be included here if the study was not published or should

be cited properly if already published. Particularly, it is not clear how many samples/probes were filtered at each step due to p-value filtering, SNP probe removal and cross-hybridising etc. Moreover, how the authors reached the final sample size n=401 and n=428 in Table S11 and Table S12?

Response: Thank you for this comment. Indeed, the DNA methylation data and methods of IBD EWAS have been published in *J Crohns Colitis* by Kalla et al.[1] To avoid overlap, we described the main methods of genome-wide DNA methylation analysis in the Supplementary Methods. The specific filter procedures and the number of samples/probes that were finally included in the IBD EWAS can be found in the work of Kalla et al.[1] We have revised the citation and added the findings of IBD EWAS in Table 4 to make this clearer. On reviewing supplementary tables, we had incorrectly added top hits for a subset of patients within this cohort and apologize for this oversight. We now include the correct supplementary tables S11 and S12 (now S7 and S8) for CD vs controls (n=449) and UC vs controls (n=456) in line with the entire cohort. Please find the details in the new table.

Methods, Line 473-477: *“As part of the IBD-CHARACTER inception cohort (EU reference no 305676), participants (154 CD patients, 161 UC patients, and 295 controls) were recruited prospectively from the gastroenterology outpatient clinic across seven centers in Europe.¹⁷”*

Supplementary Methods, Line 122-130: *“IBD patients were recruited prospectively from the gastroenterology outpatient clinic across seven centers in Europe, that were part of the IBD-CHARACTER inception cohort (EU reference no 305676).⁷ After a thorough clinical, microbiological, endoscopic, histological, and radiological evaluation, all IBD cases reached the standard diagnostic criteria. Disease clinical phenotypes (i.e., CD and UC) were diagnosed and classified according to the Lennard-Jones, Montreal and Paris criteria.⁷ People who attended the gastroenterology clinic during the same period with no symptoms and diagnosis of IBD and healthy participants were included in the control group. A total of 295 controls, 154 CD and 161 UC patients were included in this analysis.⁷”*

Supplementary Methods, Line 141-146: *“Samples and probes which did not meet the criteria for analysis were removed. For instance, samples*

containing >1% probes with detection p values >0.01 and sex mismatches were removed. Probes with bead counts of <3 in 10% of samples, or detection p values >0.01 in 10% of samples, or containing SNPs with a minor allele frequency of ≥ 0.01 in the European population in the 1000 Genomes Project were also removed.⁷

6. If I understood correctly, Table S11 and Table S12 are the DMP results for CD vs control and UC vs control, respectively; and Table 2 are the smoking-associated CpGs which showing significant associations with risk of CD or UC and were further replicated in the IBD-CHARACTER cohort (Table S11 or Table S12). However, several Table 2 CpGs were not replicated/reported in Table S11 or Table S12. Please add more details to clarify the Table 2 and its relationship to Table S11 and Table S12.

Response: Thank you for pointing this out. Your understanding of the original Table S11, Table S12, and Table 2 (now Table 3 and Table 4) is correct. Indeed, as described in Methods, we replicated the genetic loci found in epigenetic MR in the genome-wide DNA methylation analysis rather than CpG sites. This is why several CpG sites in the original Table 2 were not reported in Table S11 and Table S12. We have revised the original Table 2 (now Table 3 and Table 4), Table S11 (now Table S7), and Table S12 (now Table S8) to make it more understandable. Please find the details in the new Tables.

Table 3, Title: *“Differentially methylated genetic loci identified from the epigenetic MR of smoking in relation to the risk of Crohn’s disease and ulcerative colitis.”*

Table 4, Title: *“Validation of differentially methylated genetic loci in the genome-wide methylation analysis of Crohn’s disease and ulcerative colitis.”*

Table S7, Title: *“The differentially methylated genetic loci between CD patients and controls.”*

Table S8, Title: *“The differentially methylated genetic loci between UC patients and controls.”*

Table S9, Title: *“The differentially methylated genetic loci between CD patients and controls in whom smoking history was available, and excluding those for whom a smoking history was not taken.”*

Table S10, Title: *“The differentially methylated genetic loci between UC patients and controls in whom smoking history was available, and excluding those for whom a smoking history was not taken.”*

7. Line 250-251, how the authors come to the statement that cg03599224 and cg16280132 had a consistent effect on the regulation of gene expression from Table 2? Any other data analysis was missing here to support this result?

Response: Thank you for the comment. We would like to clarify that there are no gene expression data as part of this paper and we have altered the manuscript accordingly to reflect this. As for the regulation of DNA methylation on gene expression, we judged it according to the position of the CpG site in the corresponding gene. Generally, different locations of DNA methylation in genes have opposite effects on gene expression. When DNA methylation is located in the promoter region, it usually inhibits gene expression, whilst, when methylation is located in the gene body, it promotes gene expression.[2] Specifically, the cg03599224 and cg16280132 are located in the body and 5'UTR regions of LTA gene, respectively. Therefore, they have opposite effects on the expression of LTA gene, but their effect direction on the risk of IBD could be consistent based on the opposite beta value from epigenetic MR and EWAS. The description of these in the original manuscript may not be clear, we have revised this. Details are as follow.

Results, Line 154-159: *“In particular, altered methylation at two CpG sites located in different regions of the LTA locus, i.e., cg03599224 (located near LTA/TNF within the HLA class III locus) and cg16280132 (located at 5'UTR of LTA gene) had a consistent effect on IBD, judged by both their potential effect on gene expression and the beta value derived in epigenetic MR and IBD EWAS (Table 3 and 4).”*

8. It is highly recommended to add the methylation difference between IBD and control in Table 2 or Table S11 and S12.

Response: Many thanks for this helpful suggestion. We have revised these tables according to your recommendation. Table 2 has been reordered as Table 3 and Table 4, and its contents have also been revised. These supplementary tables contained the coefficients which represent the beta difference between

IBD and controls ($\Delta\beta$) and have been re-labelled accordingly. Please find the details in the new tables.

9. Before narrowing down the smoking-related CpG sites by the replication study, did the author explore any biological implications of the genes associated with the 46 and 69 smoking-CpGs which also show associations with the risk of CD and UC?

Response: Thank you for this comment. We have primarily focused on the CpG sites or gene loci validated in the final analysis. However, we agree with the reviewer that it may well be valuable to explore the effect of other genes on IBD risk. We have screened the published literature in PubMed to explore this and added it to Supplementary Table 4 for those genes where there may be biological relevance.

10. It is suggested that the authors add detailed legends to Figures or Tables. For instance, Figure 4 has three panels but none of them were explained in the legend. What are the headers (e.g., PP.H*) of Table S13 etc.? I would also suggest the authors to add the detailed number of mQTLs for each CpG in Table S13. Moreover, we can guess LCI and UCI in Table 1 but actually these were not indicated/used in the table, so they could be deleted in Table 1. There seems a typo for the p-value of the Crude model for CD in the last row of the Table 1.

Response: Thank you very much for your helpful suggestion. For Figure 4, each dot represents a SNP. Panel A displays the p-value of the SNP and mQTL in corresponding GWAS and EWAS. In panels B and C, the horizontal axis demonstrates the base position of the SNP and mQTL, and the vertical axis is the p-value of the SNP and mQTL, respectively.

In the colocalization analysis, the posterior probability of five hypotheses was tested: i) H0, Neither trait in this region is genetically related (PP.H0); ii) H1, Only trait 1 has a genetic association in this region (PP.H1); iii) H2, Only trait 2 has a genetic association in this region (PP.H2); iv) H3, Both traits are correlated, but the causal variables are different (PP.H3); and v) H4, Both traits are correlated and share a causal variable (PP.H4).[7] We have supplemented this in the Methods to make it easier to understand.

In Table S9, we have displayed the number of mQTLs for each CpG in the third column, which is named “n_mQTL”. To make it clearer, we have added a footnote below the Table.

In Table 1, we have deleted the words LCI and UCI and revised any errors. The details are shown in corresponding figures and tables.

Methods, Line 503-509: *“In colocalization analysis, the posterior probability of five hypotheses was tested: i) H0, Neither trait in this region is genetically related (PP.H0); ii) H1, Only trait 1 has a genetic association in this region (PP.H1); iii) H2, Only trait 2 has a genetic association in this region (PP.H2); iv) H3, Both traits are correlated, but the causal variables are different (PP.H3); and v) H4, Both traits are correlated and share a causal variable (PP.H4).⁷²”*

Figure 4, Legend: *“Figure 4. Regional plot of colocalization evidence of CpG site methylation and Crohn’s disease susceptibility. Each dot represents a SNP. The left panel displays the p-value of the SNP and mQTL in corresponding GWAS and EWAS. In the upper and low right panels, the horizontal axis demonstrates the base position of the SNP and mQTL, and the vertical axis is the p-value of the SNP and mQTL, respectively.”*

Table 1, footnote: *“HR, hazard ratio. CI, confidence interval. Numbers in bold represent significant associations.”*

11. It is very hard to read the Table S9 and Table S10, I would suggest to move them to excel tables like Table S3, S4 etc.

Response: Thank you very much. We have revised the tables according to your helpful suggestion. Their order has changed due to the new format. Please find the details in the revised supplementary tables.

12. Line 213, what is the “genetically predicted smoking behaviors”? why the “predicted” behaviors were introduced here?

Response: Thank you for pointing this out. We have revised the inappropriate wording here. Details are as follows.

Results, Line 113-115: *“We did not find any significant association between genetic predisposition to smoking behaviors and the risk of CD and UC in the main MR analysis and sensitivity analyses”*

References

1. Kalla, R., et al., *Analysis of systemic epigenetic alterations in inflammatory bowel disease: defining geographical, genetic, and immune-inflammatory influences on the circulating methylome*. J Crohns Colitis, 2022. **27**(10).
2. Suzuki, M.M. and A. Bird, *DNA methylation landscapes: provocative insights from epigenomics*. Nat Rev Genet., 2008. **9**(6): p. 465-76. doi: 10.1038/nrg2341.
3. Guida, F., et al., *Dynamics of smoking-induced genome-wide methylation changes with time since smoking cessation*. Hum Mol Genet., 2015. **24**(8): p. 2349-59. doi: 10.1093/hmg/ddu751. Epub 2015 Jan 2.
4. Joehanes, R., et al., *Epigenetic Signatures of Cigarette Smoking*. Circ Cardiovasc Genet., 2016. **9**(5): p. 436-447. doi: 10.1161/CIRCGENETICS.116.001506. Epub 2016 Sep 20.
5. de Lange, K.M., et al., *Genome-wide association study implicates immune activation of multiple integrin genes in inflammatory bowel disease*. Nat Genet., 2017. **49**(2): p. 256-261. doi: 10.1038/ng.3760. Epub 2017 Jan 9.
6. Liu, M., et al., *Association studies of up to 1.2 million individuals yield new insights into the genetic etiology of tobacco and alcohol use*. Nat Genet., 2019. **51**(2): p. 237-244. doi: 10.1038/s41588-018-0307-5. Epub 2019 Jan 14.
7. Giambartolomei, C., et al., *Bayesian test for colocalisation between pairs of genetic association studies using summary statistics*. PLoS Genet., 2014. **10**(5): p. e1004383. doi: 10.1371/journal.pgen.1004383. eCollection 2014 May.

REVIEWERS' COMMENTS

Reviewer #1 (Remarks to the Author):

The authors have responded to my previous review - I have no additional comments.

Reviewer #3 (Remarks to the Author):

Thanks for the author's efforts. The authors have addressed all of my concerns and comments.